# Poly(ADP-ribose) glycohydrolase coordinates meiotic DNA double-strand break induction and repair independent of its catalytic activity

Eva Janisiw [1,6], Marilina Raices[2], Fabiola Balmir[2,7], Luis F. Paulin [3], Antoine Baudrimont [1], Arndt von Haeseler[3,4], Judith L. Yanowitz[2], Verena Jantsch [1] & Nicola Silva [5✉]

Poly(ADP-ribosyl)ation is a reversible post-translational modification synthetized by ADP-ribose transferases and removed by poly(ADP-ribose) glycohydrolase (PARG), which plays important roles in DNA damage repair. While well-studied in somatic tissues, much less is known about poly(ADP-ribosyl)ation in the germline, where DNA double-strand breaks are introduced by a regulated program and repaired by crossover recombination to establish a tether between homologous chromosomes. The interaction between the parental chromosomes is facilitated by meiotic specific adaptation of the chromosome axes and cohesins, and reinforced by the synaptonemal complex. Here, we uncover an unexpected role for PARG in coordinating the induction of meiotic DNA breaks and their homologous recombination-mediated repair in *Caenorhabditis elegans*. PARG-1/PARG interacts with both axial and central elements of the synaptonemal complex, REC-8/Rec8 and the MRN/X complex. PARG-1 shapes the recombination landscape and reinforces the tightly regulated control of crossover numbers without requiring its catalytic activity. We unravel roles in regulating meiosis, beyond its enzymatic activity in poly(ADP-ribose) catabolism.

[1] Department of Chromosome Biology, Max Perutz Laboratories, Vienna Biocenter,  University of Vienna, Vienna, Austria. [2] Department of Obstetrics, Gynecology, and Reproductive Sciences, Magee-Womens Research Institute, University of Pittsburgh School of Medicine, Pittsburgh, PA, USA. [3] Center for Integrative Bioinformatics Vienna (CIBIV), Max Perutz Laboratories, Medical University of Vienna, Vienna BioCenter, University of Vienna, Vienna, Austria. [4] Bioinformatics and Computational Biology, Faculty of Computer Science, University of Vienna, Vienna, Austria. [5] Department of Biology, Faculty of Medicine,  Masaryk University, Brno, Czech Republic. [6] Present address: Centre for Anatomy and Cell Biology, Medical University of Vienna, Vienna, Austria. [7] Present address: AHN Center for Reproductive Medicine, AHN McCandless, Pittsburgh, PA, USA. ✉email: silva@med.muni.cz

Poly(ADP-ribosyl)ation (PARylation) is an essential post-translational modification involved in chromatin dynamics, transcriptional regulation, apoptosis, and DNA repair[1,2]. PARylation is controlled by the opposing activities of PAR polymerases, PARP1 and PARP2 (PARPs), and PAR glycohydrolase (PARG)[3,4]. The activities of PARPs are crucial for an efficient DNA damage response, as loss of PARP1 or PARP2 leads to hypersensitivity to genotoxic stress and impaired spermatogenesis in mice, while the combined deficiencies of PARP1 and PARP2 cause embryonic lethality[2,5]. Likewise, the PARG knockout is embryonic lethal in mammals and depleted cells become sensitive to ionizing radiation (IR) and show aberrant mitotic progression[1,6].

Moreover, no orthologs are present in yeast, and therefore our understanding of the roles of PARylation during germ line development has been limited. Caenorhabditis elegans (C. elegans) parg-1/PARG null mutants are viable and fertile[7,8], allowing us to analyse its function(s) during gametogenesis. It has been previously shown that parp-1/-2 and parg-1 mutants display hypersensitivity to IR exposure[7,9,10] however their roles during gametogenesis have remained poorly investigated.

In sexually reproducing species, preservation of ploidy across generations relies on meiosis, a specialized cell division program which promotes the generation of haploid germ cells[11,12]. The formation of crossovers (CO) is essential for faithful chromosome segregation into the gametes[13,14]. Connected parental homologous chromosomes (also called bivalents) can cytologically be detected in diakinesis nuclei and are thus a readout for the success of the CO establishment. COs arise by the generation and homologous recombination-mediated repair of programmed DNA double-strand breaks (DSB) effectuated by the evolutionarily conserved topoisomerase VI-like protein Spo11[15]. The activity of Spo11 is tightly regulated to ensure the correct timing, placement, and number of DSBs/COs along meiotic chromosome axes.

In C. elegans, several factors involved in promoting meiotic DSBs have been identified, and those include MRE-11, HIM-5, HIM-17, DSB-1, DSB-2, and XND-1[16–21]. Of these, XND-1 and HIM-17 are known to also influence germline chromatin structure[18,19]. DSB-1 and DSB-2 appear to have roles in maintaining DSB competency throughout early pachytene[20,21]. MRE-11 functions both in DSB formation and immediately downstream in end resection[16,22]; HIM-5 and DSB-2 have also recently been shown to couple DSB formation with HR-mediated repair[23].

Both the distribution and the abundance of DSBs and COs undergo multiple levels of regulation. In all organisms studied, the number of DSBs exceeds the number of COs, with ratios reaching 10:1 in some cases[24]. The supernumerary DSBs use HR-like mechanisms to be repaired with high fidelity, with repair intermediates shunted into non-CO (NCO) outcomes. Importantly, a robust inter-homolog repair bias ensures formation of the obligate CO in the germ cells, which in C. elegans occurs even under subthreshold levels of DSBs[17,25,26]. CO interference describes the phenomenon whereby CO-committed intermediates influence nearby DSBs to be repaired as NCOs, ensuring that COs are well-spaced across the genome[11,27]. In C. elegans, CO interference is nearly complete, as each chromosome pair receives, in most cases, only one CO[28]. On the autosomes of the worm, COs occur preferentially on the chromosome arms, away from the gene-rich region in the center of the chromosomes; while they are more evenly dispersed on the heterochromatic-like X chromosome[29].

While CO interference explains much about CO distribution in most organisms, some COs are known to arise from an interference-independent pathway. The COs generated through interference-dependent (Class I) and interference-independent mechanisms (Class II) have distinct genetic requirements, driven by MutS-MutL and Mus81 homologs respectively[30]. Genetic evidence suggests that, in C. elegans, only Class I COs are present[31,32]. Nevertheless mutants displaying interference-insensitive COs have been reported[27,33], however, these are still dependent on the canonical MSH-5/COSA-1-mediated CO pathway and they can be detected by genetic measurements of recombination[25].

CO-repair takes place in the context of the synaptonemal complex (SC), a tripartite proteinaceous structure composed of axial and central elements, arranged as a protein zipper between each pair of homologs. The SC maintains homolog associations and facilitates inter-homolog exchange of DNA during repair[34]. Crosstalk between the SC and COs is essential for modulating recombination. Incomplete synapsis dramatically weakens CO interference and additional COs per chromosome can be observed[26,35]. Conversely, reduced, but not absent, recombination levels cause premature desynapsis of the chromosome pairs that fail to establish a CO[19,36,37].

Chromosome axis components, which in C. elegans include the HORMA-domain proteins HTP-3, HTP-1/-2, and HIM-3[38–40], influence both the abundance of DSBs and the regulation of their repair.

In this study, we show an unexpected involvement of PARG-1 in influencing the dynamics of induction and repair of meiotic DSBs, and we identify a role in promoting CO formation. We found that PARG-1 functions independently of the known DSB initiation factors in efficient formation of DSBs, but it cooperates with HIM-5 to regulate global crossover numbers. PARG-1 is detected throughout the germ line and undergoes a progressive recruitment along synapsed chromosomes, culminating in the retraction to the short arm of the bivalent and enrichment at the putative CO sites. In absence of parg-1, we observe an accumulation of PAR on the meiotic chromosomes, which is suppressed by abrogation of PARP-1 and PARP-2 function. We report the association of PARG-1 with numerous key proteins composing the meiosis-specific structure of the SC both by cytological and biochemical analysis. Surprisingly, we found that PARG-1 loading, rather than its catalytic activity, is essential to exert its function during meiosis. Our data strongly suggest that PARG has scaffolding properties which are important for the fine-tuning of meiotic recombination events.

## Results

**PARG-1 is the main PAR glycohydrolase in the germ line.** The C. elegans genome encodes two orthologs of mammalian PARG, PARG-1, and PARG-2[7,8,41]. Both mutants are hypersensitive to IR exposure and more recently it was shown that parg-2 is involved in the regulation of HR-dependent repair of ectopic DSBs by influencing the extent of resection upon IR[41]. To explore possible functional links or redundancies between parg-1 and parg-2, we used CRISPR to engineer parg-2 null mutations in both the wild type (WT) and parg-1(gk120) deletion mutant backgrounds (Fig. 1a). In contrast to mammalian PARG, C. elegans parg-1 and parg-2 are largely dispensable for viability (Fig. 1b). However, abrogation of parg-1, but not parg-2 function, led to increased levels of embryonic lethality and segregation of males (which arise from X-chromosome nondisjunction[42]).

Assessment of viability, brood-size, and segregation of male progeny in parg-1 parg-2 double mutants did not reveal synthetic phenotypes but rather recapitulated the parg-1 single mutant, indicating that parg-2 does not exert prominent roles in an otherwise WT background and cannot compensate the lack of parg-1 function (Fig. 1b).

To confirm a role of PARG-1 and PARG-2 in PAR catabolism, we investigated PAR accumulation in the mutant animals.

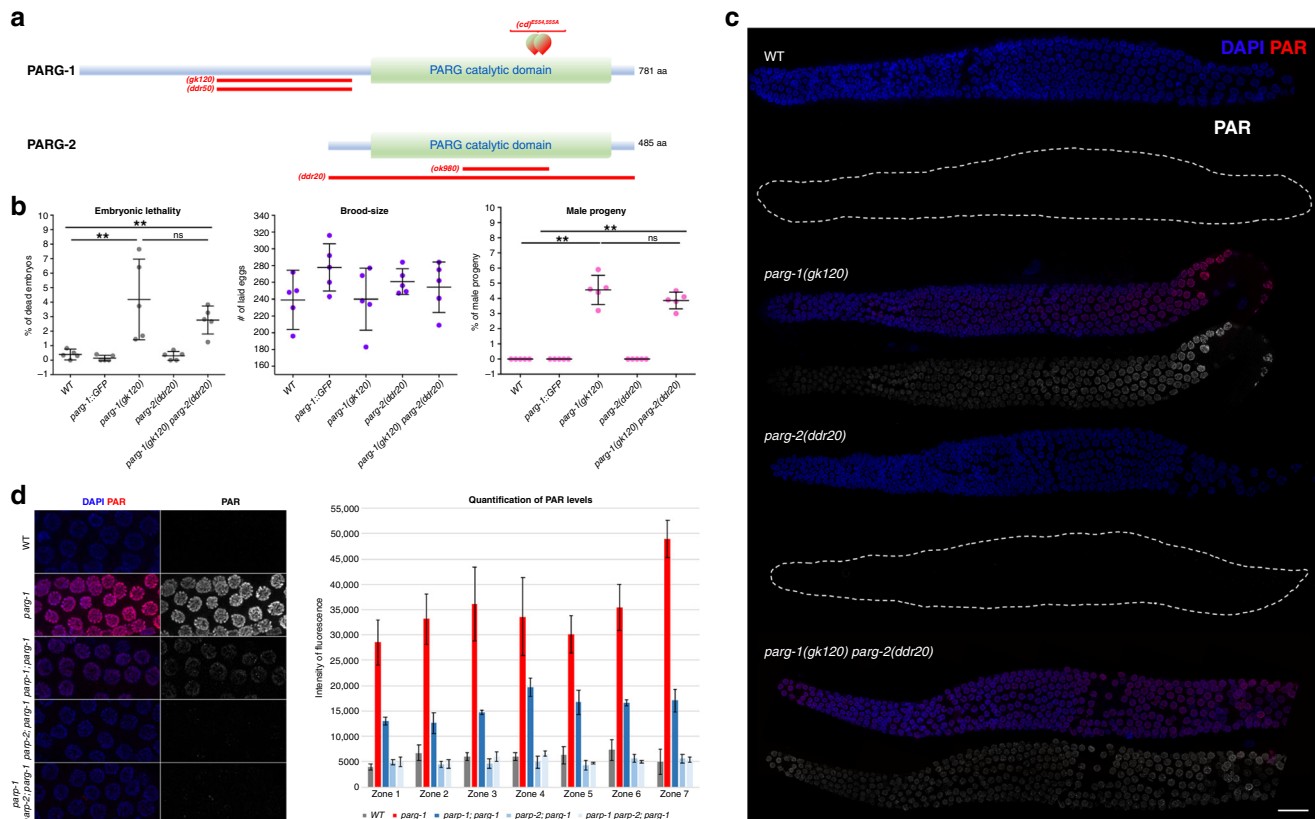

**Fig. 1 PARG-1 is the main ADP-ribose glycohydrolase in *C. elegans*. a** Schematic representation of *parg-1* and *parg-2* genetic loci. *parg-1* is predicted to encode numerous isoforms and for simplicity only isoform A is shown. Red lines or shapes delineate the position of deletions or the point mutation in the indicated mutant alleles. **b** Screening of embryonic lethality, brood-size and segregation of males in the indicated genetic backgrounds. Center of error bars indicates mean. Number of embryos scored: WT ($n = 5$, 1196), *parg-1::GFP* ($n = 5$, 1389), *parg-1(gk120)* ($n = 5$, 1200), *parg-2(ddr20)* ($n = 5$, 1304), and *parg-1(gk120) parg-2(ddr20)* ($n = 5$, 1271). Error bars indicate standard deviation, **$p < 0.0001$ assessed by unpaired *t* test and ns = not significant. **c** Representative images of whole-mount gonads from indicated genotypes showing detection of PAR by immunofluorescence. Scale bar 20 μm. Analysis was performed in biological duplicates. **d** Left: representative pictures of late pachytene nuclei from indicated genotypes showing *parp-1*- and *parp-2*-dependent accumulation of PAR in *parg-1* mutants. Gonads were divided into seven zones, encompassing the region from the distal tip cell to diplotene. Scale bar 5 μm. Right: quantification of PAR detected by immunofluorescence. Chart reports mean fluorescence intensity from at least two gonads/genotype. Error bars indicate standard deviation. Number of nuclei scored was (from zone 1 to 7): WT (97, 129, 115, 107, 102, 74, 45), *parg-1(gk120)* (129, 136, 140, 129, 130, 96, 82), *parp-1(ddr31); parg-1(gk120)* (93, 113, 123, 129, 102, 69, 44), *parp-2(ok344); parg-1* (190, 263, 215, 179, 179, 125, 72), *parp-1(ddr31); parp-2(ok344); parg-1(gk120)* (107, 140, 134, 144, 126, 103, 76).

Because PAR undergoes a rapid turnover, it cannot be detected in WT germ lines (Fig. 1c). By contrast, we detected PAR at all stages of meiotic prophase I in *parg-1* mutants. Since PAR accumulation was neither seen in *parg-2* mutants nor further enhanced in *parg-1 parg-2* (Fig. 1c), we infer that PARG-1 is the major PARG in the worm germ line.

Removal of the PAR polymerases *parp-1/-2*, suppressed accumulation of PAR in *parg-1* mutant germ cells (Fig. 1d). Interestingly, we found that while abrogation of *parp-1* function reduced PAR signal intensity to roughly 30%, lack of *parp-2* alone was sufficient to bring PAR staining to background levels. Since both *parp-1(ddr31)* and *parp-2(ok344)* mutant alleles are null, this data suggests that PARP-2 is mainly responsible for the synthesis of PAR during *C. elegans* meiotic prophase I.

Since PAR accumulates at sites of DNA damage in somatic cells[43,44], we asked whether its synthesis in meiotic prophase nuclei was dependent on the formation of meiotic DSBs. Surprisingly, we found that in the gonads of *parg-1 spo-11* double mutants, in which no programmed DSBs are made, PAR was still detectable within prophase I nuclei (Supplementary Fig. 1A), indicating that its production occurs independently of physiological DNA damage during gametogenesis.

**PARG-1 loading during meiotic prophase I requires HTP-3.** To detect PARG-1, we raised a *C. elegans*-specific anti-PARG-1 monoclonal antibody that we used in western blot analysis on both total and fractionated protein extracts (Fig. 2a and Supplementary Fig. 1B). This antibody confirmed that *parg-1(gk120)* is a null allele. We found expression of PARG-1 in both the cytosol and the nucleus in WT animals (Fig. 2a), as similarly observed in mammalian mitotic cells[45–47]. Since localization of PARG is not known in meiocytes, we employed CRISPR to tag the 3′ end of the endogenous *parg-1* locus with a GFP-tag. We assessed the functionality of the fusion protein by monitoring PAR accumulation in the gonad, embryonic lethality, and male progeny, none of which showed any differences compared to WT, indicating that PARG-1::GFP is catalytically active and fully functional (Fig. 1b and Supplementary Fig. 1A). Moreover, western blot analysis employing either anti-PARG-1 or anti-GFP antibodies on fractionated extracts from *parg-1::GFP* worms revealed identical expression as seen with untagged PARG-1 (Fig. 2b), further confirming that the GFP-tag did not affect PARG-1 stability or expression.

Immunofluorescence analyses showed that PARG-1::GFP is first detected in premeiotic and leptotene/zygotene nuclei and

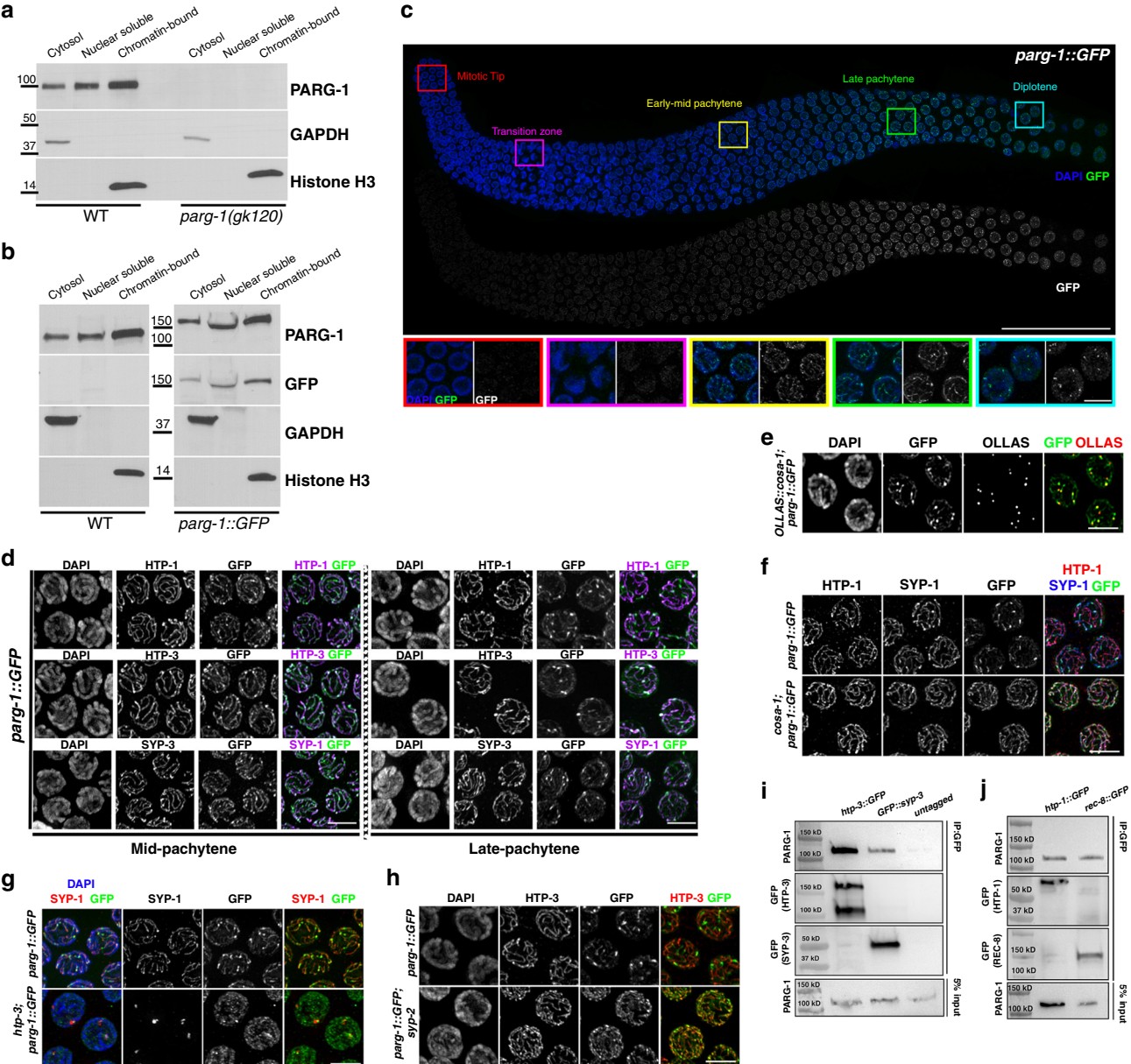

**Fig. 2 PARG-1 interacts with SC components and requires chromosome axes for proper localization. a** Western blot analysis of fractionated extracts detects PARG-1 in all subcellular compartments with enrichment in the nuclear chromatin-bound fraction. GAPDH was used as a loading control of the cytosolic fraction and Histone H3 for the chromatin-bound fraction. Analysis was performed in biological duplicates. **b** Western blot analysis of fractionated extracts showing similar expression of GFP-tagged and untagged PARG-1. Analysis was performed in biological duplicates. **c** Top: PARG-1::GFP localization in a wild-type gonad. Scale bar 30 μm. Bottom: enlarged insets showing dynamic localization of PARG-1::GFP in different stages of meiotic prophase I. Scale bar 5 μm. Analysis was performed in biological triplicates. **d** Mid- and late-pachytene nuclei of *parg-1::GFP* co-stained for lateral (HTP-1 and -3) and central component (SYP-1) of the SC. Scale bar 5 μm. Analysis was performed in biological triplicates. **e** Late pachytene nuclei showing overlapping localization of PARG-1::GFP with OLLAS::COSA-1. Scale bar 5 μm. Analysis was performed in biological triplicates. **f** Late pachytene nuclei stained for HTP-1, SYP-1, and GFP showing localization of PARG-1 along chromosomes. In *cosa-1* mutants, redistribution of PARG-1::GFP to the short arm of the bivalent is absent. Scale bar 5 μm. Analysis was performed in biological triplicates. **g** Impaired axes formation in *htp-3* mutants prevents PARG-1::GFP localization. Arrow heads indicate regions of DNA devoid of PARG-1::GFP. Scale bar 5 μm. Analysis was performed in biological triplicates. **h** PARG-1::GFP associates with HTP-3 in late-pachytene nuclei in absence of synapsis. Scale bar 5 μm. Analysis was performed in biological duplicates. **i** Western blot analysis of endogenous PARG-1 on GFP pull downs performed in *htp-3::GFP* and *GFP::syp-3* strains. Wild-type worms were used as the untagged negative control. Analysis was performed in biological duplicates. **j** Western blot analysis of endogenous PARG-1 on GFP pull downs performed in *htp-1::GFP* and *rec-8:: GFP* strains. Analysis was performed in biological duplicates.

then became progressively enriched along chromosomes throughout pachytene (Fig. 2c). In late pachytene, PARG-1::GFP showed retraction toward the short arm of the bivalent (a chromosomal subdomain formed in response to CO formation) which was particularly evident at diplotene. In nuclei at the diakinesis stage,

PARG-1::GFP was detectable mostly in the nucleoplasm (Supplementary Fig. 1C). Co-staining with axial proteins HTP-1/HTP-3 and the central SC component SYP-1[38,39,48] revealed recruitment of PARG-1::GFP onto synapsed chromosomes and confirmed its retraction to the short arm of the bivalent in late

pachytene cells (Fig. 2d), which also harbors the chiasma and the central elements of the SC[49,50]. Overlapping localization of PARG-1::GFP with both the CO-promoting factor COSA-1 and SYP-1 (Fig. 2e) further proved recruitment of PARG-1 to this chromosomal subdomain, similar to SC central elements[48,51–54]. In CO-defective *cosa-1* mutant animals, we observed that the initial loading of PARG-1::GFP to the SC was unaffected, but no retraction was observed, confirming that the redistribution of PARG-1 is dependent on bivalent formation (Fig. 2f).

Based on its localization to the SC, we tested whether PARG-1::GFP loading was dependent on chromosome axis or synapsis establishment. Loss of *htp-3*, encoding a HORMA domain-containing protein essential for axis morphogenesis[38], disrupted PARG-1::GFP localization, resulting in nucleoplasmic accumulation and occasional association with SYP-1-containing polycomplexes (Fig. 2g). By contrast, PARG-1::GFP exhibited linear staining along the chromosome axes in synapsis-deficient *syp-2* mutants (Fig. 2h), where only axial elements are loaded onto the chromosomes[34,38,39]. Thus, we conclude that PARG-1 is recruited to the SC in an HTP-3-dependent manner and its localization changes in response to CO-mediated chromosome remodeling.

Since PARG-1 localizes to chromosome axes and requires HTP-3 for loading, we wondered whether these factors formed protein complexes in vivo. To test for their possible association, we performed immunoprecipitation assays by pulling down HTP-3::GFP[55] and proceeded with western blot analysis to detect PARG-1. Robust interaction between HTP-3::GFP and PARG-1 was observed (Fig. 2i). Further, to assess whether PARG-1 establishes physical interactions with additional chromosome axis components as well, we also performed co-immunoprecipitation experiments pulling down HTP-1::GFP and REC-8::GFP[56,57]. Western blot showed that PARG-1 co-immunoprecipitated with both HTP-1 and REC-8 (Fig. 2j). Extending this analysis to the central elements of the SC component, we found that PARG-1 could be also pulled down with GFP::SYP-3[58] (Fig. 2i). Together with our localization studies, these biochemical data indicate that PARG-1 is an intrinsic component of the SC.

**PARG-1 influences processing of recombination intermediates.** Given PARG-1 recruitment along the SC and enrichment at the presumptive CO sites, we sought to investigate whether synapsis and CO formation are impaired in *parg-1* mutants. Using antibodies directed against HTP-3 and SYP-1 to monitor the establishment of the SC, we observed no difference between the WT and *parg-1* mutants (Supplementary Fig. 2A). DAPI-staining of diakinesis nuclei revealed the correct complement of six bodies as in WT worms (Supplementary Fig. 2B). Thus, we infer that *parg-1* is dispensable for synapsis and CO formation.

We next addressed whether loss of *parg-1* would impact the formation and processing of recombination intermediates by analysing the dynamic behavior of the recombinase RAD-51, which forms discrete chromatin-associated foci with a distinct kinetics of appearance and disappearance[34,59] (Fig. 3a, b). While in WT worms we see a progressive increase of RAD-51, peaking in early-mid pachytene (zone 4) and disappearing by late pachytene (zone 6), in *parg-1* mutants, we observed the delayed formation of RAD-51 foci with progressive accumulation at the pachytene stage. RAD-51 foci formation was entirely suppressed by SPO-11 removal, suggesting specific abnormalities in the induction and/or processing of meiotic DSBs rather than spontaneous or unscheduled damage arising during mitotic replication (Fig. 3a, b).

It has been previously shown that during meiosis, NHEJ is repressed in order to allow for CO repair by HR. The inappropriate activation of NHEJ at these stages can impede

RAD-51 loading[22,60–62]. Therefore, we wondered whether the delayed RAD-51 loading in *parg-1* mutants could result from NHEJ activation. To test this hypothesis, we removed the *C. elegans* ortholog of the mammalian heterodimeric KU complex subunit *cku-80/Ku80*, which is essential for NHEJ function. *cku-80; parg-1* double mutants revealed no differences in the early loading of RAD-51 compared to *parg-1* mutants, indicating that improper activation of NEHJ is likely not the cause of defect in RAD-51 loading (Supplementary Fig. 3A). However, we found a roughly 2-folds increase in the number of RAD-51 foci in late pachytene nuclei of *cku-80; parg-1* doubles compared to *parg-1 (gk120)* mutants, indicating that a fraction of recombination intermediates formed in absence of *parg-1* may be repaired by NHEJ.

Next we decided to investigate whether PARG-1 might have a role in the regulation of DSB formation. As tools to directly quantify meiotic DSBs are presently not available in *C. elegans*, we took advantage of a genetic epistasis analysis to determine if *parg-1* has a role in DSB formation. In diakinesis nuclei, DSB resection-defective mutants, such as *com-1/CtIP/Sae2* and *mre-11 (iow1)/Mre11*, display massive chromatin clumps and occasional chromosome fragments that arise from aberrant repair of meiotic DSBs. Accordingly, these clumps and fragments are fully suppressed in the DSB-devoid *spo-11* mutants[16,22,63]. Similarly, in the *com-1; parg-1* and *parg-1; mre-11(iow1)* double mutants, we found that the vast majority of diakinesis nuclei contained twelve intact univalents (Fig. 3c, d). These results are consistent with a role for PARG-1 in DSB induction but could also reflect a function for *parg-1* in targeting breaks to alternative repair pathways. To distinguish between these possibilities, we exposed the aforementioned double mutants to gamma irradiation (IR) to ectopically induce DSBs. We reasoned that if *parg-1* mutants were defective solely in DSB induction, the breaks induced by IR should restore the aberrant chromosome morphology typical of *com-1* and *mre-11*. By contrast, if *parg-1* has a role in repair pathway utilization, the IR-induced breaks would still be shunted into an alternative pathway and the appearance of DAPI bodies would remain unchanged after IR exposure. Diakinesis nuclei of irradiated *com-1; parg-1* reverted to the *com-1*-like (chromosome clumping-fusion) phenotype, supporting a putative role for PARG-1 in DSB induction. By contrast, *parg-1; mre-11(iow1)* were indistinguishable from non-irradiated controls, indicating that PARG-1 may also influence DNA repair pathway choice when *mre-11*, but not *com-1*, function is compromised (Fig. 3c, d). Together, these results suggest an involvement of PARG-1 in promoting both the formation and repair of meiotic DSBs.

**PARG-1 augments the formation of meiotic DSBs.** To further explore PARG-1's putative involvement in promoting DSBs, we tested its ability to genetically interact with mutations that are impaired in DSB induction. We combined the *parg-1(gk120)* deletion with two hypomorphic *him-17* alleles and with a *him-5* null mutation that reduce, but do not completely eliminate, SPO-11-dependent DNA breaks[17,18]. Consistent with published results, we observed that these single mutants displayed reduced numbers and delayed formation of RAD-51 foci[17,18], that was further diminished in both *parg-1; him-5* and *parg-1; him-17* double mutants (Supplementary Fig. 3B–D).

To rule out that these phenotypes could be due to secondary mutations present in the *parg-1(gk120)* mutant background, we engineered the identical deletion present in the *parg-1(gk120)* worms, giving rise to the *parg-1(ddr50)* allele (Fig. 1a). Like *parg-1(gk120)* worms, the *parg-1(ddr50)* mutants appeared to be null, as no PARG-1 protein was detected in western blot analysis carried out on total protein extracts (Supplementary Fig. 1B).

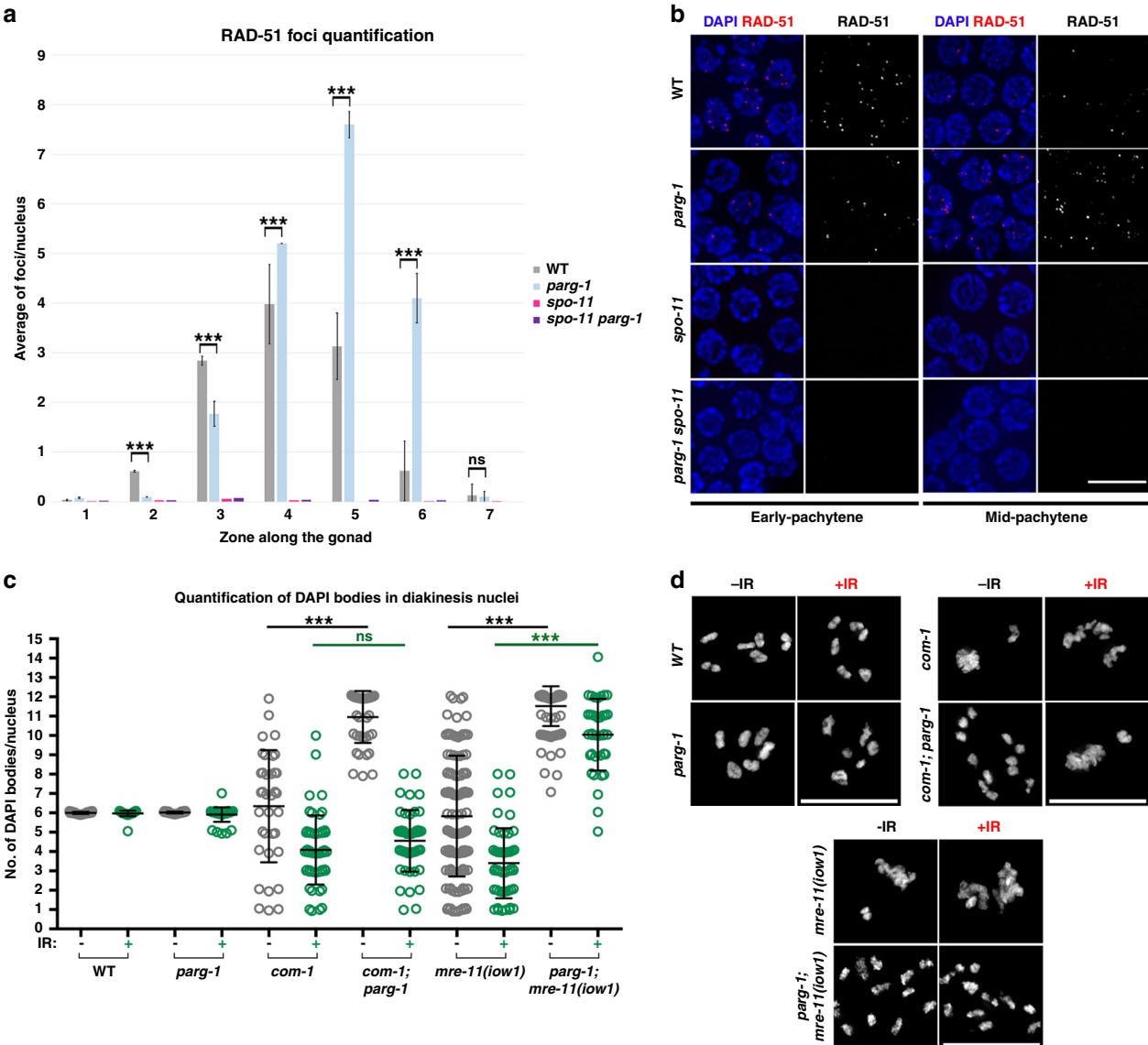

**Fig. 3 Elimination of *parg-1* function suppresses chromosome abnormalities in resection-defective mutants. a** *parg-1* mutants display SPO-11-dependent accumulation of RAD-51 foci. Error bars represent S.E.M. Asterisks indicate statistical significance assessed by unpaired *t* test (***=*p* < 0.0001 and "ns" indicates not significant). Analysis was performed in at least three gonads for each genotype. Number of nuclei analysed for each genotype (from zone 1 to 7 respectively): WT (200, 205, 127, 181, 162, 170, 90); *parg-1(gk120)* (191, 256, 248, 177, 158, 126, 135); *spo-11* (242, 236, 221, 204, 159, 158, 96); *spo-11 parg-1(gk120)* (205, 218, 203, 188, 135, 105, 81). **b** Representative examples of cells at different stages of the same genotypes analysed in (**a**). Scale bar 5 μm. **c** Analysis of diakinesis nuclei in different genotypes before and 27 h after exposure to IR. Error bars represent standard deviation. Center of error bars indicates mean. Statistical analysis was performed with two-tailed nonparametric Mann–Whitney test. (***p* < 0.0001 and ns indicates not significant differences). Number of nuclei analysed at 0 and 10 Gy respectively: WT (35–94), *parg-1(gk120)* (33–88), *com-1* (37–55), *com-1; parg-1* (42–48), *mre-11* (124–52), *parg-1; mre-11* (110–40). **d** Representative images of diakinesis nuclei of the same genotypes scored in (**c**). Scale bar 5 μm.

Quantification of RAD-51 foci numbers in *parg-1(ddr50)* and *parg-1(ddr50); him-5* mutants was comparable to *parg-1(gk120)* and *parg-1(gk120); him-5*, respectively, indicating that the aberrant RAD-51 expression profile specifically arises from *parg-1* loss of function rather than unrelated mutations (Supplementary Fig. 3E). The defects in RAD-51 filament formation observed in the *parg-1; him-17* and *parg-1; him-5* double mutants were correlated with defective loading of pro-CO factors such as HA::RMH-1, GFP::MSH-5 and OLLAS::COSA-1 (Fig. 4b, c and Supplementary Fig. 4) as we would expect for mutations that impair DSB formation. Analysis of diakinesis nuclei revealed an extensive lack of chiasmata (Fig. 4d) and enhanced embryonic lethality (Supplementary Fig. 3F) in the double mutants due to defects in CO repair. These phenotypes

were observed for both *parg-1(gk120)* and *parg-1(ddr50)* alleles in combination with *him-5* mutation, further ruling out possible involvement of secondary mutations.

Loading of RMH-1, MSH-5, and COSA-1, as well as bivalent formation, were largely, although not completely, rescued by IR exposure (Fig. 4b–d and Supplementary Fig. 4), further corroborating that the lack of COs was due to impaired DSB formation. Abrogation of PARG-1 function also exacerbated the CO defect observed in both young (day #1) and old (day #2) *dsb-2* mutants (Supplementary Fig. 3G), which display an age-dependent loss in the proficiency to induce DSBs[21]. These results indicate that loss of *parg-1* function impairs a parallel, *him-17-*, *him-5*, and *dsb-2*-independent pathway for DSB induction.

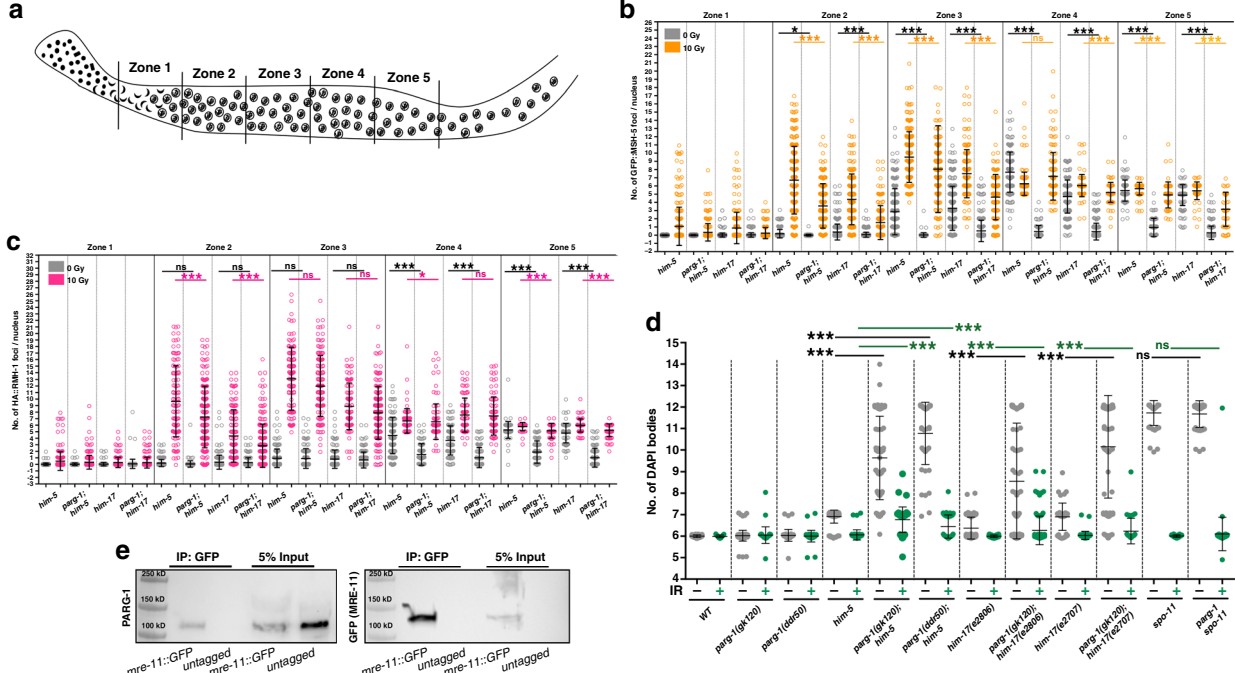

**Fig. 4 PARG-1 promotes formation of meiotic DSBs and interacts with MRE-11. a** Schematic representation of the gonad divided into five equal regions spanning transition zone throughout late pachytene, employed for MSH-5 and RMH-1 foci quantification. **b** Quantification of GFP::MSH-5 foci in the indicated genotypes before and after IR. Center of error bars indicates mean. Error bars indicate standard deviation and asterisks indicate statistical significance assessed by unpaired t test (***p < 0.0001, ns = not significant). Number of nuclei analysed is reported in Supplementary Table 1. **c** Quantification of HA::RMH-1 foci in the indicated genotypes before and after IR. Center of error bars indicates mean. Error bars indicate standard deviation and asterisks indicate statistical significance assessed by unpaired t test (***p < 0.0001, ns = not significant). Number of nuclei analysed is reported in Supplementary Table 1. **d** Quantification of DAPI-bodies of different genotypes before and after IR exposure. Center of error bars indicates mean. Error bars indicate standard deviation and asterisks indicate statistical significance assessed by unpaired t test (***p < 0.0001, ns = not significant). Number of nuclei analysed is reported in Supplementary Table 2. **e** Western blot analysis of endogenous PARG-1 on GFP pull downs performed in mre-11::GFP and untagged wildtype strains (negative control). Analysis was performed in biological duplicates.

To quantify DSBs that progress to strand exchange intermediates, we took advantage of the *rad-54* mutation in which removal of RAD-51 from D-loops cannot occur properly and the RAD-51 foci that accumulate are thought to reflect the total number of DSBs that are made[26,64,65]. We generated the *rad-54; parg-1* double mutants and analysed RAD-51 dynamics. Strikingly, the number of RAD-51-labeled recombination intermediates was greatly reduced in the double mutant (Supplementary Fig. 3H). We also found that a large number of diakinesis nuclei contained normal appearing DAPI-bodies (Supplementary Fig. 3I), in stark contrast to *rad-54*, where chromosomes morphology is highly aberrant[23]. RAD-54 absence both impairs RAD-51 turnover and prolongs the "window of opportunity" during which DSBs can be made[20,21,26], therefore the phenotypes in the *rad-54; parg-1* double mutants can be explained as a consequence of reduced DSBs but also as an alternative form of repair, further reinforcing a possible dual role for PARG-1 in the formation and the processing of the recombination intermediates.

**PARG-1 interacts with MRE-11.** To further interrogate PARG-1 function in DSB formation, we next sought to investigate the interplay between PARG-1 and DSB-promoting factors. To this end, we assessed the localization of the pro-DSB factors HIM-5::3xHA, HIM-17::3xHA, DSB-2, and XND-1 in *parg-1* mutants. We observed no gross defects in localization compared to the controls (Supplementary Fig. 5A–D), which suggests that PARG-1 is not required for the loading of these pro-DSB factors. Conversely, PARG-1::GFP loading appeared normal in *him-5, dsb-2,* and *him-17* (null and hypomorph alleles) mutant backgrounds.

The only difference compared to WT is the lack of retraction of PARG-1::GFP to the short arm of the bivalent, which is a consequence of the lack of COs caused by these mutations (Supplementary Fig. 5E) similar to *cosa-1* mutations (described above). Given the synergistic phenotypes observed in the double mutants and the lack of defects in the loading/expression of DSB-promoting proteins, we conclude that PARG-1 supports formation of DSB via alternative pathway(s) to the known pro-DSB factors in *C. elegans*.

It has been previously shown that DSB formation in worms is also promoted by the axial component HTP-3, possibly through its interaction with the MRN/X complex factor MRE-11, known to be involved in meiotic break induction[22,38,66]. Since we already showed an interaction between HTP-3::GFP and PARG-1 (Fig. 2i), we now wanted to address if this extended to an association with MRE-11. Western blot analysis for PARG-1 on GFP pull downs performed with the *mre-11::GFP* transgene[67] also showed co-immunoprecipitation (Fig. 4e). This suggests that the PARG-1-mediated activity in promoting meiotic DSBs may intersect the HTP-3-MRE-11 axis.

**PARG-1 and HIM-5 modulate crossover numbers.** While the loading of pro-CO factors was largely rescued in the irradiated *parg-1; him-5* double mutants, over half of the diakinesis nuclei still displayed univalents (Fig. 4d), indicating substantial, yet incomplete, restoration of chiasmata. The dose employed in our irradiation experiments (10 Gy) sufficed to fully elicit bivalent formation in *him-5, spo-11, parg-1 spo-11* (Fig. 4d), and *spo-11; him-5*[68]. Therefore, we conclude that additional CO execution

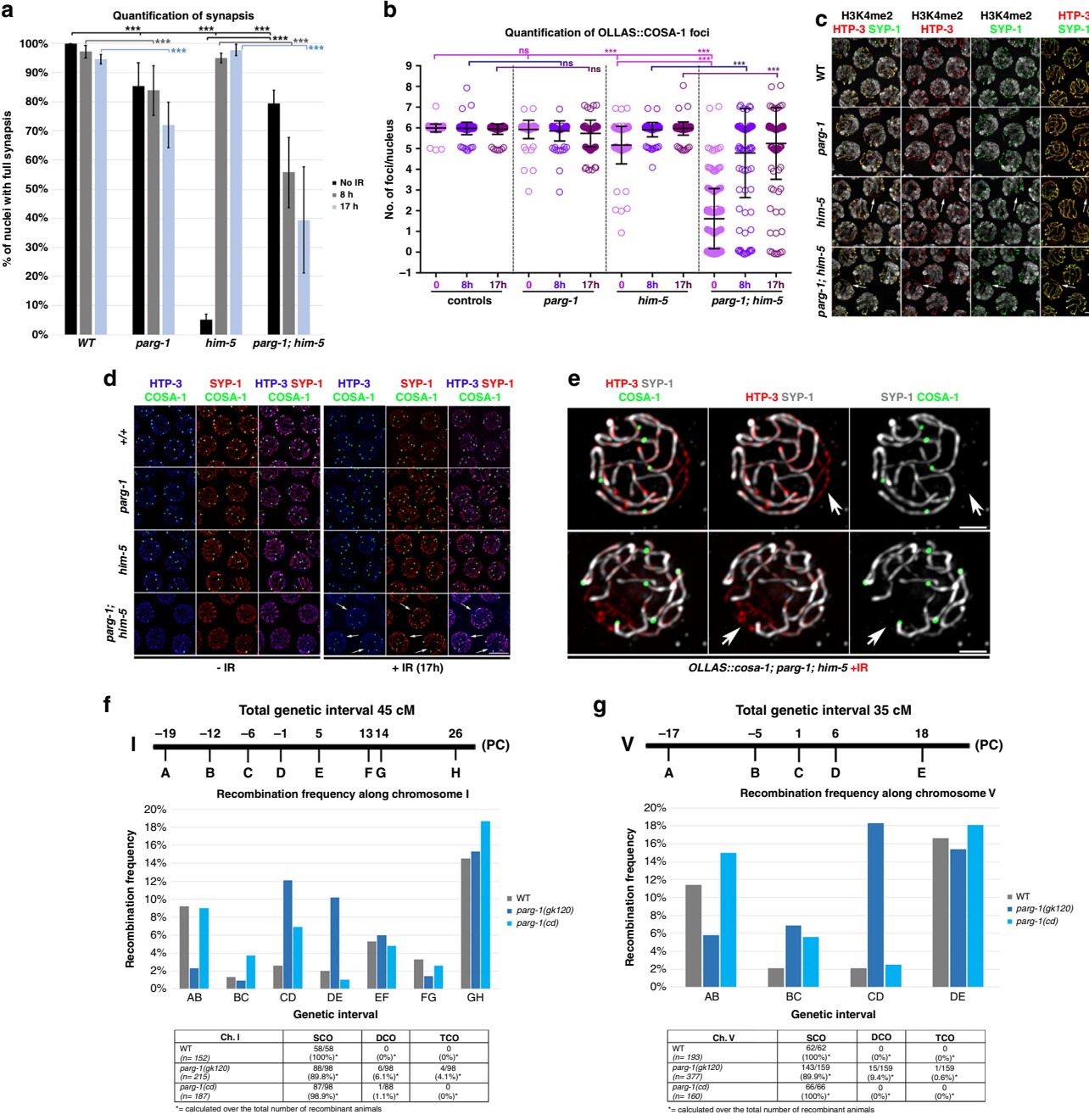

steps are defective in *parg-1; him-5*. Importantly, this phenotype was not observed in *parg-2; him-5* double mutants, in which the number and structure of DAPI bodies resembled *him-5* single mutants both before and after exposure to IR (Supplementary Fig. 6). To assess whether *parg-2* provided activity in absence of *parg-1*, we generated the *parg-1 parg-2; him-5* triple mutants and analysed diakinesis nuclei before and after exposure to IR, which did not differ from the *parg-1; him-5* double mutant (Supplementary Fig. 6). This further corroborates that the recombination defects observed in *parg-1; him-5* are a consequence of impaired *parg-1* function and that *parg-2* does not play significant roles under contemporary *him-5* deficiency.

The SC is a dynamic structure that responds to the presence or absence of (as yet unidentified) CO intermediates in the nucleus. When COs are made, they stabilize the SC in *cis*[36,37,69]. In genetic backgrounds with reduced DSB induction, such as those described above, the chromosome pairs lacking a CO undergo

desynapsis at a late pachytene stage, whereas in mutants that completely lack COs, homologs remain fully synapsed, but the SC subunits are more labile[36,37]. Given both the partial rescue of chiasmata formation in *parg-1; him-5* double mutants after IR and also the localization of PARG-1 to the SC, we sought to determine if CO designation and SC dynamics are decoupled by simultaneous loss of both HIM-5 and PARG-1 functions.

In unirradiated *him-5* mutant worms, the sole absence of a CO on chromosome X caused its extensive desynapsis in late pachytene nuclei (Fig. 5a, c), recapitulating previous observations[37]. By contrast, in *parg-1; him-5* double mutants, the majority of nuclei showed full synapsis (Fig. 5a–c), in agreement with the fact that desynapsis is not triggered when CO establishment is fully abrogated[36,37]. In support of this interpretation, we show that nuclei containing fully synapsed chromosomes displayed no COSA-1 loading in unirradiated *parg-1; him-5* double mutants (Fig. 5d). Immunostaining for H3K4me2, a histone modification that

**Fig. 5 PARG-1 and HIM-5 regulate CO numbers. a** Quantification of synapsis in late pachytene nuclei without IR and at different times after IR exposure, by SYP-1 and HTP-3 co-staining. Only nuclei showing complete colocalization of HTP-3 and SYP-1 were considered fully synapsed. Quantification was performed in the last seven rows of nuclei before entering the diplotene stage. Error bars indicate S.E.M. and asterisks indicate statistical significance assessed by $\chi^2$ test (significance level for $p < 0.05$; ***$p < 0.0001$, ns = not significant). Number of nuclei analysed is reported in Supplementary Table 3. **b** Quantification of OLLAS::COSA-1 foci formation in late pachytene nuclei in the same genotypes and at the same time points as in (**a**) under physiological conditions of growth and at different times post exposure to ionizing radiations. Center of error bars indicates mean. Error bars indicate standard deviation and asterisks indicate statistical significance assessed by unpaired $t$ test (***$p < 0.0001$, ns = not significant). Number of nuclei analysed is reported in Supplementary Table 3. **c** Immunostaining of H3K4me2, HTP-3, and SYP-1 to assess chromosome X synapsis in different genotypes. Scale bar 5 μm. Analysis was performed in biological duplicates. **d** Co-staining of OLLAS::COSA-1 with SYP-1 and HTP-3 shows desynapsis associated with lack of CO but normal numbers of COSA-1 foci on the remaining chromatin in *parg-1; him-5* double mutants. Scale bar 5 μm. Arrows indicate examples of unsynapsed regions in nuclei containing six COSA-1 foci. Analysis was performed in biological duplicates. **e** High magnification of late pachytene *parg-1; him-5* nuclei after 8 h (top) and 17 h (bottom) post irradiation, showing normal numbers of COSA-1 foci despite desynapsis. Arrows indicate desynapsed chromosome regions (presence of HTP-3, absence of SYP-1). Scale bar 1 μm. Analysis was performed in biological duplicates. **f** Top: schematic representation of the genetic position of the SNPs employed to assess the recombination frequency on chromosome I. PC indicates the position of the pairing center. Middle: recombination frequencies assessed in each of the genetic intervals in wild type, *parg-1(gk120)* and *parg-1(cd)* mutants. Bottom: table displaying number and percentage of single, double and triple crossovers (SCO, DCO, and TCO respectively) in the analysed genotypes. *n* indicates number of worms analysed. **g** Same analysis as in (**a**), performed for chromosome V. Statistical significance was calculated by $\chi^2$ test considering the number of expected and the number of observed recombinant animals for each chromosome and analysis is reported in the Supplementary Table 4, which shows that recombination frequency in *parg-1(gk120)* is statistically different from WT worms, while in *parg-1(cd)* it is not different from controls (significance level for $p < 0.01$).

shows specific enrichment on the autosomes, but not on the X chromosome[70], further revealed that the X chromosome was fully synapsed in *parg-1; him-5* doubles, consistent with the lack of a CO, and in stark contrast to *him-5* or *parg-1* single mutants (Fig. 5c).

We next wanted to address whether SC stabilization and CO formation are coordinated in the *parg-1; him-5* double mutants after irradiation, where six COSA-1 foci were observed (Fig. 5b) but univalents resulted (Fig. 4d). For this analysis, we undertook a time course analysis, examining pachytene nuclei 8 and 17 h after irradiation. Six COSA-1 foci were observed in *parg-1* mutants both before and after IR; however, we found a mild, albeit statistically significant, reduction in the number of nuclei with full synapsis (Fig. 5a), suggesting that PARG-1 might exert roles in promoting efficient establishment or stabilization of the SC. In the *him-5* single mutant, 10 Gy of IR is sufficient to both rescue COSA-1 loading and to suppress X-chromosome desynapsis, as observed both 8 and 17 h post-IR, as shown previously[37].

In *parg-1; him-5*, COSA-1 foci numbers were also largely rescued at 8 h post-IR and remained steady at 17 h post-IR (Fig. 5b). However, desynapsis was observed at 8 h post-IR and synapsis was further reduced 17 h after irradiation (Fig. 5a). Strikingly, a substantial number of nuclei exhibited desynapsis, yet showed the full complement of six COSA-1 foci (8 h = 52% and 17 h = 74.3%) (Fig. 5d, e), a situation never described in other meiotic mutants. COSA-1 foci were never associated with unsynapsed regions. The fact that these nuclei contained six COSA-1 foci, as in WT animals, suggests that some chromosomes bear additional COSA-1 marked CO events. These results revealed that the global regulation of CO-mediated repair is profoundly perturbed in the absence of PARG-1 and HIM-5 functions.

To further characterize the defects in *parg-1; him-5* mutants, we examined the meiotic progression marker phospho-SUN-1$^{S8}$[71]. In WT animals, SUN-1$^{S8}$ is phosphorylated in leptotene/zygotene and dissipates at mid-pachytene[72]. The lack of DSBs or impaired homologous recombination-mediated repair trigger retention of phospho-SUN-1$^{S8}$ at the nuclear envelope until the late pachytene stage[72]. In DSB-defective mutants, but not in mutants with impaired recombination (such as *cosa-1*), delayed removal of phospho-SUN-1$^{S8}$ is rescued by exogenous DSB induction[20,21,37,72]. Since *parg-1; him-5* double mutants appear to carry defects in both DSB induction and repair, we analysed phospho-SUN-1$^{S8}$ localization before and after IR exposure to assess whether these phenotypes could be uncoupled by phospho-SUN-1$^{S8}$ dynamics.

*parg-1* mutants displayed mild prolongation of phospho-SUN-1$^{S8}$ staining (Supplementary Fig. 7), consistent with the delayed accumulation of RAD-51 foci (Fig. 3a and Supplementary Fig. 3). *him-5* and *parg-1; him-5* mutants showed comparable, prolonged phospho-SUN-1$^{S8}$ staining under unchallenged growth conditions, consistent with defective DSB induction and recombination. While IR exposure fully suppressed the persistence of phospho-SUN-1$^{S8}$ in the *him-5* as expected, it only mildly suppressed it in *parg-1; him-5* (Supplementary Fig. 7). The inability of IR to suppress phospho-SUN-1$^{S8}$ accumulation further reinforces the conclusion that lack of both PARG-1 and HIM-5 impairs both meiotic DSB formation and repair.

**PARG-1 shapes the recombination landscape.** Given the involvement of *parg-1* in regulating not only DSB formation, but also homology-mediated repair, we investigated the recombination frequency in different genetic intervals on chromosome I and V by monitoring SNP markers in Bristol/Hawaiian hybrids, which allowed us to assess both CO numbers and their position[73]. We found a striking increase of COs in the central regions of both chromosomes (Fig. 5f, g), where COs are usually absent in the WT[74]. In addition, double and triple COs were observed, albeit at a low frequency. These results revealed that impaired *parg-1* function impacts the global levels and distribution of COs and weakens CO interference in *C. elegans*.

**PARG-1 acts independently of its catalytic activity.** We next sought to investigate whether PARG-1 catalytic activity is necessary to exert its function during meiosis. To this end, we generated a *parg-1* "catalytic-dead" mutant (referred to as *parg-1 (cd)* hereafter) using CRISPR to mutate two glutamates in the catalytic domain (E554, 555A). These amino acids are conserved throughout evolution and were shown to be essential for PARG activity in vitro in both mammals and nematodes[7,44,75]. Immunostaining analysis in *parg-1(cd)* and *parg-1(cd)::GFP* revealed accumulation of PAR on meiotic chromosome axes as in *parg-1 (gk120)* null mutants, indicating that also in vivo E554-E555 are necessary for PAR removal (Fig. 6a). Western blot analysis showed that the overall levels of both PARG-1$^{CD}$::GFP and untagged PARG-1$^{CD}$ were increased, ruling out possible artefacts due to the addition of GFP (Fig. 6b). The blots were also probed with anti-PAR antibodies and this confirmed that both strains

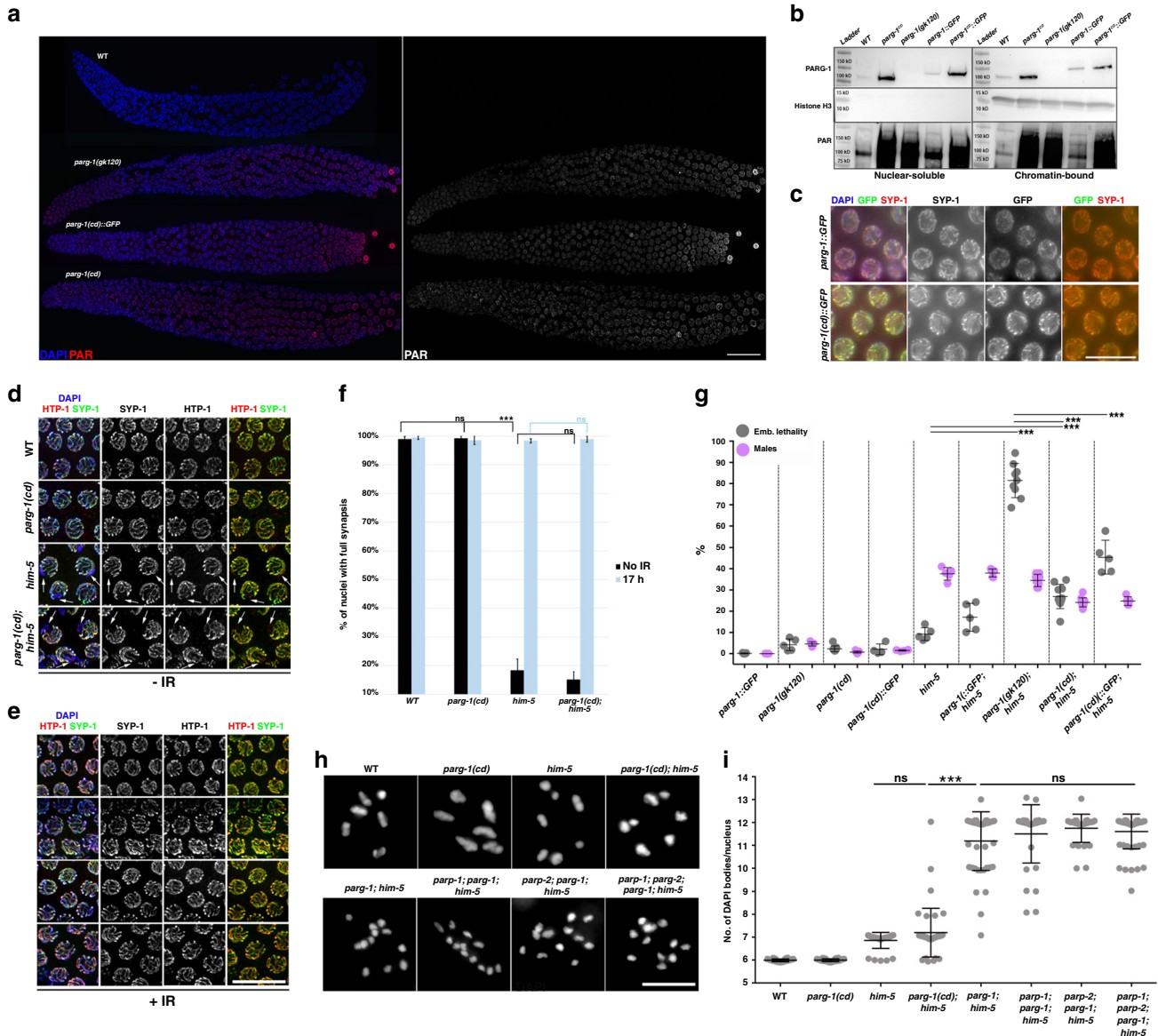

**Fig. 6 PARG-1 catalytic activity is dispensable for recombination. a** Left: whole-mount germlines of the indicated genotypes stained with anti-PAR antibodies and DAPI. Right: PAR staining. Scale bar 20 μm. Analysis was performed in biological duplicates. **b** Western blot with fractionated extracts shows higher PARG-1CD::GFP and PARG-1CD abundance compared to controls. Anti-histone H3 was used as loading control for chromatin-bound fraction. Western blot (bottom) confirmed accumulation of PAR in both *parg-1(cd)* and *parg-1(gk120)* null mutants. Analysis was performed in biological duplicates. **c** Non-deconvolved images of late pachytene nuclei showing SYP-1 and elevated levels of PARG-1CD::GFP versus PARG-1::GFP with delayed redistribution in late pachytene nuclei. Scale bar 10 μm. Analysis was performed in biological duplicates. **d** The X-chromosome undergoes desynapsis in *parg-1(cd); him-5* double mutants. Arrows indicate the unsynapsed X chromosome in the indicated genotypes before irradiation. Scale bar 10 μm. Analysis was performed in biological duplicates. **e** Exposure to IR fully rescues X-chromosome desynapsis in *parg-1(cd); him-5* double mutants. Scale bar 10 μm. Analysis was performed in biological duplicates. **f** Quantification of synapsis by SYP-1 and HTP-3 co-staining in the indicated genotypes before and after IR exposure. Error bars indicated S.E.M. Statistical significance was assessed by $\chi^2$ test (significance level for $p < 0.05$, ***$p < 0.0001$, ns = not significant). Number of nuclei analysed (0 Gy-10 Gy): WT (215–205), *parg-1(cd)* (137–146), *him-5* (169–190), *parg-1(cd); him-5* (165–212). **g** Blocking PARG-1 catalytic activity caused milder synergistic effects when combined with *him-5* mutants in contrast to *parg-1(gk120); him-5*. Center of error bars indicates mean. Error bars indicate standard deviation and asterisks show statistical significance calculated by two-tailed nonparametric Mann–Whitney test. (***$p < 0.0001$). Number of animals scored: *parg-1::GFP* (5), *parg-1(gk120)* (5), *parg-1(cd)::GFP* (4), *him-5* (5), *parg-1::GFP; him-5* (5), *parg-1(gk120); him-5* (9), *parg-1(cd); him-5* (10), *parg-1(cd)::GFP; him-5* (10). **h** DAPI-staining and quantification of DAPI-bodies (**i**) in diakinesis nuclei for the indicated genotypes. Center of error bars indicates mean. Error bars indicate standard deviation and asterisks show statistical significance calculated by two-tailed nonparametric Mann–Whitney test. (***$p < 0.0001$). Scale bar 5 μm. Number of nuclei analysed is reported in Supplementary Table 2.

have compromised glycohydrolase activity. PARG-1CD::GFP was expressed and loaded in meiocytes (Fig. 6c) but displayed prolonged localization along the chromosomes in late pachytene cells, were PARG-1 normally is retained mostly at the short arm of the bivalent in control animals (Fig. 2).

To assess whether the catalytic activity of PARG-1 was required for the induction and/or repair of meiotic DSBs, we analysed the *parg-1(cd); him-5* double mutants as described above.

X-chromosome desynapsis (Fig. 6d–f) was suppressed upon IR exposure in both *parg-1(cd); him-5* and *him-5* mutants, in

contrast to the desynapsis seen in *parg-1(gk120); him-5* after IR (Fig. 5). Offspring viability was only mildly reduced compared to *him-5* mutants (Fig. 6g). This indicates robust fidelity of chromosome segregation in contrast to the *parg-1(gk120); him-5* double mutants. Moreover, analysis of recombination frequency in the *parg-1(cd)* revealed a recombination landscape that was similar to control animals (Fig. 5f, g), in contrast to the central shift observed in *parg-1(gk120)* nulls, further corroborating that the catalytic activity of PARG-1 is largely dispensable to regulate recombination. These results suggest that PARG-1 loading onto chromosomes and/or a non-catalytic function of PARG-1 are essential to avert recombination defects in the absence of HIM-5. This interpretation was further reinforced by the observation that the simultaneous removal of *parp-1* and *parp-2* did not rescue CO formation in *parg-1(gk120); him-5* mutants (Fig. 6h, i), indicating that CO defects are independent of PAR. Thus, we conclude that the glycohydrolase activity of PARG-1 is not required to promote induction of meiotic DSBs and their homologous recombination-mediated repair.

## Discussion

PARylation has been extensively studied in the context of the DNA damage response in mitotic mammalian cells, where it facilitates the repair of DNA lesions by promoting both the recruitment of repair factors and mediating local chromatin relaxation around damage sites[76–79]. In contrast to PARP1/2, the functions of PARG have been much less investigated due to the lack of a suitable model system, since PARG null mutants are embryonic lethal in mammals[4]. We found that the *C. elegans* PARG-1 regulates DSB induction, in parallel to the so far known HIM-17/HIM-5/DSB-1/DSB-2-dependent routes. Moreover, our data demonstrate that PARG-1 regulates homology-directed repair of DSBs by operating within a functional module with HIM-5 to ensure the efficient conversion of recombination intermediates into post-recombination products, ultimately controlling global CO numbers.

Our cytological analysis, in combination with co-immunoprecipitation assays (Fig. 2), identified PARG-1 as an intrinsic component of the SC, where it is recruited via interaction with the chromosome axis protein HTP-3. Studies in mammalian mitotic cells reported nucleoplasmic localization of PARG and robust recruitment onto the DNA lesions induced by laser microirradiation[43,44]. The association with a meiosis-specific structure such as the SC therefore suggests distinct functional regulation in meiotic cells. Interestingly, PARG-1 retracts to the short arm of the bivalent and becomes enriched with SYP proteins at the presumptive CO sites in late pachytene nuclei (Fig. 2c, d), a localization also described for DNA repair and CO-promoting factors[51–54]. Nevertheless, abrogation of synapsis did not impair loading of PARG-1 along the chromosomes, a prerogative typically observed for axial rather than central components of the SC[38–40,49]. This would suggest that PARG-1 may be targeted to both lateral and central elements of the SC or shift from the former to the latter upon CO-mediated chromosome remodeling.

In support of a dynamic model of PARG-1 localization, PARG-1 was found in protein complexes both with HTP-1, HTP-3, and REC-8, all proteins localizing to chromosome axes[38,50,80], and also with SYP-3, which is a component of the central part of the SC[81]. We believe that the localization of PARG-1 to the chromosome axes and its interaction with HTP-3 might hold crucial functional implications for promoting formation of meiotic DSBs and/or affecting their repair outcome. Many axial proteins, including *C. elegans* HTP-3, have been shown to directly influence abundance of DSBs during meiosis in several organisms[38,82], while others, such as HTP-1 and HIM-3[39,40], have been directly

involved in modulating repair. Therefore PARG-1 might exert its pro-DSB functions by operating from within the SC.

An activity of PARG-1 in promoting meiotic DSB formation by directly regulating pro-DSB factors is less likely, since the synergistic effects between *parg-1* and *him-17-him-5-dsb-2* mutants (Fig. 4, Supplementary Fig. 5) clearly place PARG-1 in a parallel, distinct pathway. Consistently, expression and localization of PARG-1 and HIM-17, HIM-5, or DSB-2 were not mutually dependent (Supplementary Fig. 5). We cannot rule out the possibility that PARG-1 may contribute to DSB formation by modulating SPO-11 activity or its recruitment to the presumptive DNA break sites, which we could not address due to unavailability of tools for SPO-11 analysis in worms.

An additional argument in support of a model where interaction with HTP-3 might be key for PARG-1-mediated pro-DSB function, comes from its co-immunoprecipitation with MRE-11 (Fig. 4e), a proven interaction partner of HTP-3[38]. MRE-11 holds important roles in break resection across species and in *C. elegans* also in break formation[22,83]. MRE-11 has been invoked as a putative substrate intersected by HTP-3 function in inducing meiotic breaks[38]. Therefore, PARG-1 might act together with HTP-3 and MRE-11 to ensure normal levels of breaks. The fact that *parg-1; mre-11* animals displayed such different phenotypes compared to *com-1; parg-1* further highlights a complex, yet undetermined, functional interaction between these two factors that might impact more than only DSB formation. As abrogation of *parg-1* activity suppresses chromosome fusions triggered by aberrant resection or by impaired *rad-54* function, a regulatory activity exerted by PARG-1 on DNA repair pathway choice is a possible scenario.

The generation of unstructured chromosome masses in diakinesis nuclei in both *mre-11(iow1)* and *rad-54* mutants can be suppressed by PARG-1 removal and leads to the formation of achiasmatic chromosomes. This would be consistent with a repair switch toward the sister chromatid, thus PARG-1 might selectively direct DSB repair towards the homologous chromosome. PARG-1 might operate through its localization along the SC, which has been shown to strongly influence meiotic DNA repair. More experimental analysis will be necessary to unravel the roles of PARG-1 during repair.

Our analysis also revealed that PARG-1, both independently and in combination with HIM-5, plays important roles in the global regulation of meiotic recombination. In fact, *parg-1* mutants show a profoundly perturbed recombination landscape, as distribution of COs displayed a marked shift towards the center of the autosomes (Fig. 5), a chromosome domain normally devoid of COs in WT animals[29]. This feature has also been observed in mutants with reduced levels of bivalent formation or aberrant DSB repair[17,53,84–86]. Moreover, CO interference appeared weakened in absence of *parg-1*, suggesting a diminished stringency in the control of CO numbers.

The intermediates formed upon abrogation of *parg-1* function are nonetheless fully competent to be processed as COs, as long as HIM-5 function is preserved. In fact, while bivalent formation was fully restored in *parg-1; him-17* double mutants upon IR exposure (Fig. 4d) (arguing for a rescue of reduced DSB levels), diakinesis of irradiated *parg-1; him-5* mutant worms showed only a partial restoration of chiasmata, highlighting a repair defect as well (Fig. 4d). The mutual requirement of PARG-1 and HIM-5 in the reciprocal mutant background suggests the presence of a repair mechanism that relies on these two factors in order to efficiently complete inter-homolog recombination repair. Both *him-5* and *dsb-2* exert regulatory functions on DNA repair pathway choice during gametogenesis[23] and our work also highlights *parg-1* as an important factor operating within such a process.

Simultaneous abrogation of *parg-1* and *him-5* function caused much more severe aberrations than just reduced recombination: we found that IR exposure restored COSA-1 loading to the WT levels (six foci/nucleus) in pachytene cells (consistent with impaired break formation); nevertheless large portions of chromatin were devoid of SYP-1/COSA-1 in many of these nuclei. These data indicate that additional COs have been designated on remaining, SC-associated chromosomes (Fig. 5).

Previous studies in ex vivo somatic cells suggested possible functions of PARG that are independent of its catalytic activity or PAR synthesis[44]. Our data show that in catalytically impaired *parg-1(cd)* mutants, which consistently accumulate PAR as in *parg-1(gk120)* nulls (Fig. 6), the inactive protein was recruited at higher levels and displayed delayed redistribution along the chromosomes in late pachytene nuclei. This is in agreement with reports in mammalian cells showing that PARG$^{KD}$ is recruited to laser-induced microirradiation sites with faster kinetics compared to PARG$^{WT}$ and that this recruitment is only partially dependent on the PARP1 function[44]. Strikingly, PARG-1$^{CD}$::GFP was still capable of promoting chiasmata formation on the autosomes in *him-5* mutants: in fact, the embryonic viability and numbers of DAPI-bodies in *parg-1(cd)*; *him-5* were comparable to *him-5* single mutants before and after IR exposure, and importantly, desynapsis was not observed. This suggests that the loading of PARG-1, rather than its enzymatic activity for PAR removal, was sufficient to induce DSBs and promote efficient bivalent formation in the presence of exogenous DSBs. This was further corroborated by the fact that in the *parp-1*; *parp-2*; *parg-1 (gk120)*; *him-5* quadruple mutants, bivalent formation was not rescued, demonstrating that the roles exerted by PARG-1 in promoting DSB induction and meiotic repair are independent of PARylation. In mammalian mitotic cells, it has been shown that PARG interacts with PCNA through a non-canonical PIP-box, and mutations in this domain, while (i) abrogating interaction with PCNA, (ii) preventing PARG recruitment at damage sites, (iii) as well as its localization in the replication foci, they do not perturb catalytic activity; conversely, mutations in the PARG catalytic domain do not affect interaction with PCNA[43]. This is consistent with our findings that PARG holds important roles in cellular homeostasis that are independent of its enzymatic function, highlighting the multifaceted nature of this protein.

Our data further demonstrate that the catalytic activity and the scaffolding properties of PARG are required for distinct cellular processes (Fig. 7). Our study establishes a crucial role of PARG during meiotic prophase I in augmenting induction of meiotic DSBs and regulating their repair via HR in a metazoan model. Further studies are necessary to clarify whether PARG-1 recruitment affects the structure of the SC resulting in the modulation of DSB formation and recombination, or whether the presence of PARG-1 along the chromosomes influences the recruitment and dynamic behavior of other factors, which ultimately exert a regulatory role in DSB formation and recombination. Our work highlights the multifaceted aspects of PARG in vivo not simply as an enzyme mediating the catabolism of PAR, but also as a pivotal factor intersecting multiple functional branches acting during meiosis.

## Methods

**Genetics**. Worms were cultured at 20 °C according to standard conditions. The N2 strain was used as the WT control. We did not notice any significant differences between *him-17(e2707)* and *him-17(e2806)* alleles and the former has been employed for the majority of the experiments unless otherwise indicated. The *parp-1(ddr31)* is a full knockout allele that we generated by CRISPR. In most of the experiments, the *parg-1(gk120)* allele was employed unless otherwise indicated in brackets. All the strains employed for this study are reported in the Supplementary Table 6.

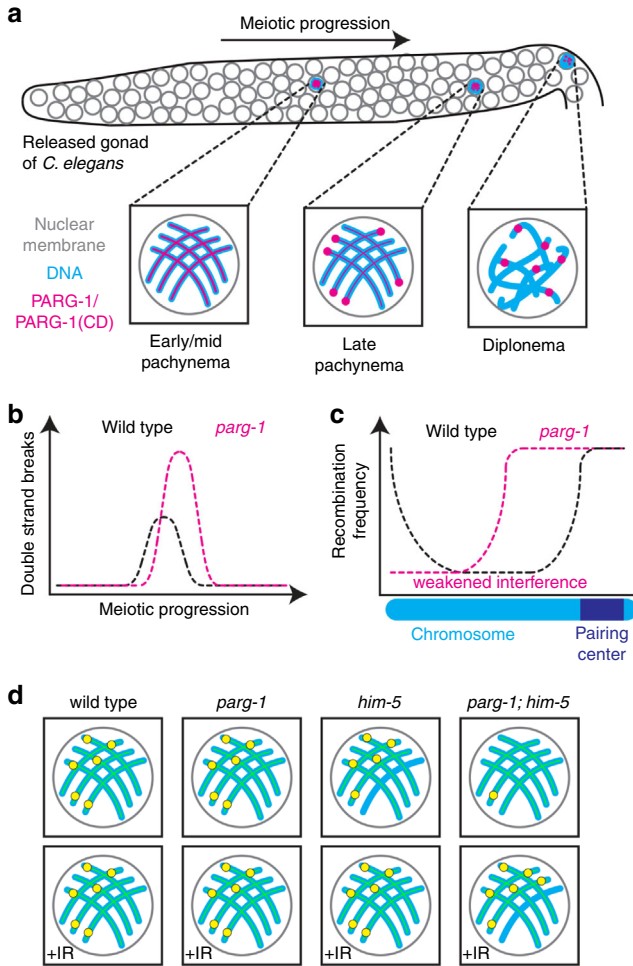

**Fig. 7 Graphic schematization of possible PARG-1 modes of action. a** PARG-1 associates with the SC through its binding with HTP-3 independent of its enzymatic proficiency. **b** Loading of PARG-1 is necessary for efficient formation of recombination intermediates, and **c** to shape the recombination landscape, to prevent unscheduled CO occurrence and weakening of CO interference. **d** Through cooperation with HIM-5, PARG-1 ensures normal levels of DSBs, and their legitimate repair via HR by influencing the global numbers of inter-homologue recombination intermediates and their chromosome-wide location. This ultimately ensures maintenance of synapsis.

**Screenings and RNA interference**. L4 worms were individually plated and moved onto fresh plates every 12 h for 3 days. Dead eggs were scored 24 h after the mother had been removed and male progeny after 3 days. Embryonic lethality and male progeny were calculated as the fraction of unhatched eggs/total laid eggs and males/total hatched eggs respectively.

RNA interference for *syp-1* was performed by employing the clone from the Ahringer library. Bacteria were streaked on agarose plates containing 12.5 mg/ml of tetracycline and 100 mg/ml of ampicillin. Single colonies were grown in 20 ml of LB with 100 mg/ml of ampicillin at 37 °C overnight and the following day, bacteria were spun at maximum speed for 20 min and the pellet was resuspended in 2 ml of LB with 100 mg/ml of ampicillin. 100 μl of concentrated bacteria were spotted on NG Agar plates containing 100 mg/ml of ampicillin and 1 mM IPTG and transcription of RNA was induced at 37 °C overnight. The following day, WT L3 worms were placed on the induced plates and let laying eggs for 3 days, after which, the mothers were removed. L3 F1 worms were picked onto freshly induced plates and dissected as young adults (24 h post-L4 stage).

**Antibodies**. The following antibodies at the indicated dilutions were employed for immunolocalization studies: rabbit polyclonal anti-HA (SIGMA, 1:1000), rabbit polyclonal anti-OLLAS (Genscript, 1:1500), rabbit polyclonal anti-PAR (Trevigen, 1:1000), mouse monoclonal anti-GFP (Roche, 1:500), guinea pig polyclonal anti-HTP-3 (1:500)[38], guinea pig polyclonal anti-HTP-3 (1:750) (Y. Kim lab), chicken

polyclonal anti-SYP-1 (1:500)[56], rabbit polyclonal anti-SYP-1 (this study, 1:1000), rabbit polyclonal anti-HTP-1 (1:500)[50], rabbit polyclonal anti-RAD-51 (Novus, 1:10,000), guinea pig polyclonal anti-phospho-SUN-1$^{S8}$ (1:750)[72], rabbit polyclonal anti-DSB-2 (1:5000)[21], guinea pig polyclonal anti-XND-1 (1:2500)[19], and mouse monoclonal anti-H3K4me2 (Millipore, 1:250). All the secondary antibodies were Alexafluor-conjugated and used at 1:300.

The following antibodies at the indicated dilutions were employed in western blot analysis: rabbit polyclonal anti-HA (SIGMA, 1:3000), mouse monoclonal anti-HA (Cell Signaling, 1:1000), mouse monoclonal anti-PARG-1 (this study, 1:500), chicken polyclonal anti-GFP (Abcam, 1:4000), rabbit polyclonal anti Histone-H3 (Abcam, 1:100,000), goat polyclonal anti-actin (Santa Cruz, 1:3000), mouse monoclonal anti-Tubulin (Thermofisher, 1:2000), and mouse monoclonal anti-GAPDH (Ambion, 1:5000). HRP-conjugated secondary antibodies were purchased from Thermofisher and were used at 1:10,000 (goat anti-chicken), 1:15,000 (goat anti-rabbit) and 1:8000 (goat anti-mouse).

**Cytological procedures and image acquisition**. For immunostaining experiments, synchronized worms of the indicated age were dissected and processed as previously described[52] except for detection of PARG-1::GFP and GFP::MSH-5. Briefly, worms were dissected in PBS and immediately placed in liquid nitrogen. Slides were placed in cold methanol at −20 °C for 1′ and then fixed with 2% PFA in 0.1 M $K_2HPO_4$ (pH 7.4) for 10′ in a humid chamber at room temperature. Samples were then processed as for regular staining. For GFP::MSH-5 detection, worms were dissected and fixed in 2.5% PFA for 2′ at room temperature and then freeze-cracked in liquid nitrogen. Slides were placed in absolute ethanol at −20 °C for 10′ and then washed in 1× PBST. DAPI staining was performed as for normal staining and GFP was directly acquired without employing a primary anti-GFP antibody. For quantification of PAR (Fig. 1d), samples were acquired with identical settings and equally adjusted in Fiji. Gonads were divided into seven equal regions from mitotic tip to diplotene entry and a circle of fixed area was employed to assess absolute fluorescence in each nucleus with Fiji as in ref. [52]. For quantification of RAD-51 foci, gonads were divided into seven equal regions from the mitotic tip to the diplotene entry and number of RAD-51 foci was counted in each nucleus. Number of nuclei analysed is reported in the Supplementary Table 5. Quantification of phospho-SUN-1$^{S8}$ extension was performed as in[87].

Most images were captured using a Delta Vision system equipped with an Olympus IX-71 microscope and a Roper CoolSNAP HQ2 camera with Z-stack set at 0.25 μm of thickness. Images in Fig. 6d, e were acquired with a Delta Vision system equipped with an Evolve 512 EMCCD Camera; images in Figs. 6c, h and S8, were acquired with an upright fluorescence microscope Zeiss AxioImager.Z2 equipped with a Hamamatsu ORCA Flash 4.0, sCMOS sensor camera, using UPlanSApo 100×/1.4 Oil objective. All images were deconvolved using Softworx (Applied Precision) except for images in Fig. 6c, h, which are non-deconvolved, and Supplementary Fig. 8, where deconvolution was performed with ZEN 3.0 Blue software (Zeiss), using "constrained iterative" algorithm at maximum strength. Images were analysed in Photoshop, were some false coloring was applied.

**Biochemistry**. Fractionated protein extracts were produced as previously described[56] and co-immunoprecipitation assays and Western Blot were performed as previously shown[52]. At least 500 μg of nuclear extract (pooled nuclear-soluble and chromatin-bound fractions) were used for IPs and 30 μg of each fraction were used for Western blot of fractionated extracts. Agarose GFP-traps (Chromotek) were employed for pull downs following manufacturer instructions. Buffer D (20 mM HEPES pH 7.9, 150 mM KCl, 20% glycerol, 0.2 mM EDTA, 0.2% Triton X-100 and complete protease inhibitor) was used for incubation with beads and washes.

For western blot on whole-cell extracts, 200 synchronized young adults were hand-picked into 32 μl of 1 × TE buffer containing 1× Protease inhibitor cocktail (Roche), flash-frozen in liquid nitrogen, and Laemmli buffer to 1× final concentration was added after defrosting. Worms were boiled for 10′ and then extracts were run on a precast 4–20% gradient acrylamide gel (Biorad).

**Generation of PARG-1 and SYP-1 antibodies**. To generate the mouse monoclonal anti-PARG-1(2D4) antibody, the cDNA encoding for residues 1-350 of *C. elegans* PARG-1 (isoform A) was generated by gene synthesis (IDT) and then cloned into pCoofy31 in frame with a C-ter 6 × His tail. The resulting plasmid was expressed in *E. coli* BL21 cells according to standard procedures and 1 mg of purified protein was used to immunize three mice in the "in-house" monoclonal antibody facility at Max Perutz Laboratories. Raw sera were screened by western blot employing extracts produced from WT, *parg-1(gk120)* and *parg-1::GFP* worms in order to identify immunoreactive bands against PARG-1. Spleen cells from one mouse were fused with myeloma cells to generate hybridoma cell lines and mixed clones were successively diluted to gain monoclonal line 2D4, from which the antibody was harvested. Antibody specificity was assessed by Western blot, where an immune reactive band of the expected MW of approximately 90 kDa in WT but not in *parg-1* mutant worms was detected (Fig. 2a).

A synthetic peptide corresponding to amino acids 2-24 of SYP-1 protein (DNFTIWVDAPTEALIETPVDDQS) was used to generate anti-SYP-1 polyclonal antibodies in rabbits (Genscript). Raw sera were affinity purified and employed for immunostaining analysis. Anti-SYP-1 antibody was tested by immunofluorescence,

where it robustly detected SYP-1 in WT worms whereas the signal was largely gone upon *syp-1$^{RNAi}$* (Supplementary Fig. 8).

**Irradiation**. Irradiation assays were performed as previously described[52]. For quantification of synapsis and OLLAS::COSA-1 foci number in late pachytene nuclei, worms were dissected at the indicated time after irradiation and quantification was performed in the last seven rows of nuclei before diplotene entry. For quantification of HA::RMH-1 and GFP::MSH-5 in Fig. 4, worms were dissected 8 h post-IR and gonads from transition zone to late pachytene were divided into five equal regions and number of foci/nucleus was assessed. For diakinesis analysis, worms were dissected 24–27 h post irradiation. The dose employed for all irradiation experiments was 10 Gy. Number of nuclei analysed for each condition are reported in Supplementary Table 1.

**CRISPR-Cas9 genome editing**. Generation of tagged or mutated lines was performed as previously described[52]. Briefly, to tag endogenous *parg-1* locus, GFP was amplified by PCR with primers carrying 25 base pairs of homology to the left and right side of the STOP codon of *parg-1* gene. To generate the PARG-1$^{E554,555A}$ catalytic-dead mutant, a synthetic ultramer (IDT) was employed, in which we included silent mutations to produce an Alu I restriction site for screening purposes. The mutations were generated in both WT and *parg-1::GFP* genetic backgrounds. To elicit a full knockout of *parp-1*, we employed two sgRNAs targeting the beginning and the end of the gene. The *him-17::3xHA* and *him-5::3xHA* were generated by employing synthetic DNA ultramers (IDT) and N2 worms were used. All the tagged lines carried a 5x-Gly linker between the tag and the coding region. The *parg-1(ddr50)* line carries the same deletion present in the VC130 strain, which we generated in both WT and CB4856 strains, by employing a synthetic ultramer (IDT). All the lines generated by CRISPR were outcrossed to WT worms at least twice before use.

**Recombination assay**. The recombination landscape was assessed following the same strategy as in[28], by exploiting different Dra I digestion pattern of SNPs present in the Bristol and Hawaiian genetic backgrounds. Briefly, *parg-1(gk120)* and *parg-1(cd)* mutations were generated by CRISPR in both the N2 (Bristol) and CB4856 (Hawaii) strains. Bristol/Hawaiian F1 hermaphrodite hybrids carrying the indicated mutations were backcrossed to Bristol males carrying a tdTomato fluorescent reporter expressed in the soma in order to monitor recombination frequency in the oocytes. The relevant regions containing the SNPs for chromosomes I (snp_F56C11, snp_Y71G12, pkP1052, snp_DI007, snp_F58D5, CEI-247, uCEI-1361, snp_Y105E8B)[88] and V (pkP5076, snp_Y61A9L, pkP5097, R10D12, snp_Y17D7B)[88] in the indicated genetic intervals were amplified by PCR and the products digested with Dra I to monitor recombination patterns. Data presented in Fig. 5 refer to the total number of worms analysed in independent replicates. Statistical analysis of the differences in the recombination rate for each genetic interval between the mutant backgrounds and the controls is reported in the Supplementary Table S4.

**Reporting summary**. Further information on research design is available in the Nature Research Reporting Summary linked to this article.

## Data availability
All data generated or analysed during this study are included in this published article (and its Supplementary Information files). The Source data underlying Figs. 1b, d, 3a, c, 4b–d, 5a, b, f, g, 6f, g, i and Supplementary Figs. 2B, 3A–I, 4A, B, 5A, 6A, 7A are provided as Supplementary Information. All data are available from the authors upon reasonable request. Source data are provided with this paper.

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

## Acknowledgements
We are grateful to E. Martinez-Perez, M. Zetka, A. Villeneuve, Y. Kim, and S. Smolikove for strains and reagents; A. Graf for performing the microinjections; D. Slade for helpful comments and discussion throughout the development of this work; L. Krejčí, S. Uldrijan, and D. Šmajs for sharing equipment. Some strains were provided by the CGC, which is funded by NIH Office of Research Infrastructure Programs (P40 OD010440). N.S. was funded by an Interdisciplinary Cancer Research (INDICAR) fellowship by the Mahlke-Obermann Stiftung and the European Union's Seventh Framework Program for Research, Technological Development under grant agreement no 609431; by the Grant Agency of Czech Republic (GA20-08819S) and a "Start-Up" grant from the Department of Biology of Masaryk University. V.J. lab receives funding by the Austrian Science Fund (FWF; project no. P-31275-B28); A.v.H. lab by DK RNA (UW: W1207) and FWF URSPRUNG 2018 (I-1824-B22); J.L.Y. lab by NIGMS (2 R01 GM104007); M.R. was funded in part by an MWRI postdoctoral fellowship. We acknowledge the core facility CELLIM supported by the Czech-BioImaging large RI project (LM2018129 funded by MEYS CR) for their support with obtaining scientific data presented in this paper.

## Author contributions
N.S. designed the research and performed most of the experiments with the technical support of E.J.; M.R., F.B., and J.L.Y. generated some strains, analysed diakinesis chromosomes, performed the recombination assay on chromosome I, and carried out statistical analysis of the data; L.F.P. and A.v.H. analysed whole genome sequencing data, which initiated the analysis of the catalytic-dead *parg-1* mutants; A.B. produced the *HA::rmh-1* tagged line and made Fig. 7; V.J. provided logistic, infrastructure, resources, and conceptual support; J.L.Y., V.J., and N.S. wrote the paper.

## Competing interests
The authors declare no competing interests.
