## [Peer Review File · Nature Communications]

Reviewers' Comments:

Reviewer #1:

Remarks to the Author:

The manuscript by Janisiw et al., describes the role of poly(ADP-ribose) glycohydrolase (PARG-1) in the *C. elegans* germ line. Using a combination of genetic, cell biological and biochemical approaches the authors find that PARG-1 localizes to and physically interacts with axial element and central region components of the SC, and recombination proteins, and concentrates to the short arm of the bivalent in response to crossovers. The authors provide evidence that PARG-1 functions in DSB formation, influences repair outcomes and alters the crossover landscape. Interestingly, the catalytic activity of this enzyme is not required for the reported meiotic functions, leading to a model whereby PARG-1 is functioning as a scaffold to facilitate different aspects of meiotic recombination. In general the study is well done and will be of interest to a broad audience. However, the reader is left with a vague understanding of what PARG-1 is doing to promote these different events. This may be corrected by inclusion of additional experiments and in the writing.

1. Abstract: This is set up for elucidating the role of ADP ribosylation; however, the findings indicate that PARG-1's function in meiotic processes are independent of its catalytic activity. Please consider rewriting with less of a focus on ADP ribosylation.
2. In many places in the text the authors state that PARG-1 functions independently of known DSB initiation factors yet cooperates with HIM-5 (one of those initiation factors). This seems at odds to me (e.g., lines 120-121). I think all of the data are consistent with PARG-1 functioning in a parallel pathway (and perhaps not in DSB formation, see 4).
3. Figure 2: Please define CY, NS, CB directly on figure.
4. PARG-1's role in DSB formation: The authors provide several indirect pieces of evidence that PARG-1 plays a role in DSB formation. However, there is also evidence that it does not affect DSB formation. Although RAD-51 foci formation is delayed in the *parg-1* mutant, the numbers of RAD-51 foci are actually higher than in wild type. What does RAD-51 look like following IR treatment in the *parg-1* mutant? Are early RAD-51 foci restored? Inactivation of *rad-54*, which traps RAD-51 on the filament, has been used as a proxy to determine total number of RAD-51-loaded breaks, examining RAD-51 in *parg-1*; *rad-54* mutants may help clarify the role of *parg-1* in DSB formation. Further the finding that *mre-11(iow1)*; *parg-1* mutants still have 12 univalents following IR treatment but *com-1*; *parg-1* do not is not well explained. Interestingly, *mre-11(iow1)*; *cku-80* mutants have delayed RAD-51 foci formation while *com-1*; *cku-80* do not, providing parallels with the *parg-1* mutant. What does RAD-51 look like in the *mre-11(iow1)*; *parg-1* and *com-1*; *parg-1* double mutants in the presence and absence of IR? What about *parg-1*; *cku-80*? At a minimum, these findings need to be discussed more to help the reader understand the complicated genetic interactions reported in the context of the published literature.
5. Figure 4: Please indicate significance of the most important interactions directly on B and C.
6. Figure 5: As chromosomes were not traced, please tone down the conclusion that an entire chromosome is desynapsed and therefore lacking a COSA-1 focus.
7. Figure 6: Please provide more details on how COs were examined. Were DCOs and TCOs confirmed with independent SNPs? Please provide statistical analyses of the data. I also think it is important to examine COs in the *parg-1*; *him-5* double mutant, the basis for uncovering the role in COs. In the materials and methods, the authors indicate that *parg-1(cd)* was generated in Hawaiian and Bristol implying that COs were analyzed, but no data on this mutant is included in Figure 6.
8. The authors should consider including a model, which may help them frame the discussion, and guide the reader in understanding PARG-1's role in the germ line.

Reviewer #2:

Remarks to the Author:

I realise it is a custom to briefly summarise the main findings of the paper in a review rapport but

the abstract suffices for this purpose as it is a perfect (and precise) reflection of the main findings of the study and its novelty.

To me this is really an excellent paper! First of all, the technical quality of the experimental work is outstanding and state-of-the-art (an example being the generation and analysis of tagged and altered endogenous alleles), and the depth of the study is also impressive, using the full spectrum of tools available to the *C. elegans* research field. Second, I found the paper very well written, and importantly: all conclusions supported by presented data. Finally, the paper present novel biology that is relevant to the broader community: it reveals a new, physiologically very meaningful, role for PARG in meiosis, in particular, in the regulation of crossover formation, making optimal use of the fact that *C. elegans* tolerate complete lack of PARG activity

I have very little to comment or suggest to improve the quality of the paper. As a minor issue I guess I would transfer the germline PAR staining in the *parg-1(cd)* animals from the supplement to the main figure, as I found it more convincing than (or at least importantly additive to) the Western depicted Fig 7B.

I was also very much intrigued about the puzzling observation that IR could not induce the aberrant chromosome morphology typical of *com-1* and *mre-11* single mutants in the *mre-11 parg-1* double (in marked contrast to that in *com-1 parg-1* doubles), and without more experimental support or indications to what PARG is doing, I would be a bit more reserved than stating in the discussion section that this data is "indicating that PARG-1 can act as a switch in channelling DSB repair into multiple branches."

Overall, I found the paper a great read and my compliments to the authors for such great work.

Reviewer #3:

Remarks to the Author:

PARylation is a protein modification used in DNA damage repair. However, little is known about it in meiosis, where DNA damage is naturally formed. Much less known about the role of PARylation removal in meiosis. In *C. elegans*, there are two PAR glycohydrolase: PARG-1 and PARG-2. The manuscript by Janisiw et al is focused on the role PARG-1 plays in the *C. elegans* germline, as PARG-2 doesn't seem to play a role in the germline under normal conditions. PARG-1 localizes, and physically interacts, with the synaptonemal complex. *parg-1* mutants show a delay and a small accumulation of RAD-51 foci. Although crossover numbers are not affected in the *parg-1* mutant, the *parg-1* mutant act synergistically with mutants with partial defects in crossover formation and suppress defects in recombination of mutants that target repair to NHEJ. Most of the defects observed in the double mutants can be suppressed by introducing DSBs using irradiation. These findings are interpreted as a role for PARG-1 in DSB formation. The contribution of PARG-1 (by its own) is small - embryonic viability is not much effected and the obligatory crossovers still form. All of these phenotypes are independent of the catalytic activity of PARG-1. The *parg-1* mutant also shows altered distribution of crossovers and decrease in interference, indicating additional function for PARG-1 in crossover regulation. Overall, the discovery that PARG-1 plays a role in meiotic recombination and that this function is independent of its catalytic domain is novel and exciting. The analysis of the *parg-1* mutant phenotypes is very nicely and thoroughly done and the quality of the imaging performed is excellent. If the conclusion stated in this paper will be supported by addressing the concerns below, it will likely provide findings that will be of interest to the community.

1) In *parg-1* mutants RAD-51 foci delay in their appearance and peak later than in wild type. *parg-1* mutants can enhance the phenotype of mutants that affect the formation of DSBs, and this effect can be partially suppressed by irradiation. These observations can be interpreted in two ways: 1) PARG-1 has an effect on the timing of DSB formation: in its absence DSB formation is

delayed, 2) PARG-1 is promoting both DSB formation and DSB repair: in its absence less DSBs are formed but the DSBs formed take longer to repair. The authors favor option #2. However, option #1 is just as likely and is more parsimonious. DSB formed later (compared to wild type) may be more sensitive to perturbation of the DSB formation machinery explaining their synergistic effect with mutants such as *him-17* or *dsb-2*. DSB formed later may be formed in a germline region that is less permissive to NHEJ, explaining why *parg-1* can suppress chromosomal fusions in *com-1* or *mre-11* mutants. The partial suppression by IR can be explained by the timing of IR breaks: IR breaks are induced throughout the germline and likely the diakinesis oocytes scored originated from breaks in the distal germline. Therefore, by exposing to IR breaks are added before where they are made in *parg-1* mutants, suppressing the DSB timing defects. Unless there is a very good reason to disregard model #1 the papers should be written in a way that accepts both possibilities equally, including the discussion, title and abstract.

2) I find the fact that the catalytically dead mutants share no phenotypes with the null mutant interesting but also concerning. A major takeaway message of this paper is that PARGs can play a non-catalytic role in DSB formation and CO regulation. Since this is such an unexpected result, it needs to be reinforced. Both null and CD mutations increase PAR, therefore they are targeting PARG-1, but it is possible that an additional mutation in the null background is responsible for the null phenotypes (even if this allele was outcrossed X6, linked mutations may still be present). It will be better to confirm some basic phenotypes of the null with more than one allele. CRISPR/Cas9 is widely used in *C. elegans* and in the hands of the authors and deletion alleles can be generated in a span of few weeks. It will be advisable to generate a full null allele of *parg-1*. There is no need to examine all phenotypes with this allele, but to examine the phenotypes that are different between the CD and the null alleles (EMB, anti-RAD-51, and if possible, the interaction with *him-5*).

3) Statistical analysis is missing for much of the data. In some cases, there are error bars but not statistics. It is essential to perform this analysis (Fig 1B, 4B, 3A, 4B, 4C, 4D, 5A, 5B, 6A, 6B, 7C, 7E, 2SB, S3B, S3C, S5A, and S7A).

4) Others (Bae et al 2020 FASEB letters) have shown a role for PARG-2 in the germline in response to IR. Here the paper shows that PARG-2 plays no role in the germline, but this is done in the absence of IR. One way to reconcile both observations is for *parg-2* gene/PARG activity to be induced by IR. If so, PARG-2 expression could compensate for the lack of *parg-1* activity in IR experiments, suppressing the defects observed in the absence of IR. In this case, the suppression of the phenotype of double mutant (with *parg-1*) by IR can be attributed to PARG-2 activation. This should be tested (for example, by staining for PAR in germline of *parg-1* mutants a few hours after IR - is PAR staining weaker compared with no IR?).

5) Most of the figures are written in a small font that makes it hard to examine them. The font size varies between panels. Using a larger and standard font size can improve the figures.

6) The western blot showing that there is no PARG-1 protein in the mutant is shown just for 100kDa. However, if residual protein was made it should have been smaller (since the mutant contains large deletion), so the WB that includes the expected size following this deletion should also be shown. The fact that *gk120* is not a complete deletion opens the door for gain of function/hypomorphic phenotypes, so it's important to completely rule this out.

7) What is the localization pattern of PARG-1 in diakinesis? The data we see is from late pachytene, but a clearer image should be observed in diakinesis.

8) It is a little bit perplexing that PARG-1 co-IPs with every single protein tested. All these IPs use a GFP tagged protein. Could that create a problem?

9) I find it confusing that the same type of data is presented in different ways. For example, could

the data pending DAPI body number (3C, 4D, 7F), embryonic lethality (1B, 2C) presented in the same way?

10) How many repeats/replicas were there for the western blot/IP data?

11) PARG-1 in 2C runs at 100KDa, while in other westerns it runs higher- is this a typo?

12) Line 173 states that PRAG-1 is expressed in all nuclear compartments, but the figure legend says it is enriched in the nucleus, which fits the figure better. Which one is right and how reproducible?

13) Based on 2G, It's hard to see that the localization is in the nucleoplasm. It looks like it's still associated with chromosomes, but since the chromosomes are not synapsed it doesn't look like wild type localization. A zoom-in with an arrow showing GFP where there is no DAPI will be helpful.

14) 2I- why 2 bands with HTP-3?

15) 4B and 4C are very hard to read. I suggest focusing on one or two zones and showing the rest in sup figure.

16) 5A and B- why are there less COSA-1 foci and more desynapsis in *parg-1* single mutant following IR? Statistics will help here...

17) Figure 5 and 6 can be fused to one figure.

18) If a point wants to be made about increased localization in CD mutants, 7A should be quantified (like in 2D).

19) Line 500-503 "*parg-1(cd)* mutants...displayed delayed redistribution along the chromosomes in late pachytene" I cannot see that.

20) Line 703-707: indicate which data was collected from 2 vs. 3 germlines. 2 germlines may be too little for some readers.

21) Image 1A- please include scale bar in bp. It will also be useful to have a cartoon of the proteins with domain structure, what the deletions remove and where is the CD mutations. This is important since the catalytic domain is examined. It will also be informative to discuss the structure of PARG-1 and 2 and their similarities to PARGs found in other organisms, which will be important for readers outside the *C. elegans* field.

22) Discussion: the discussion could benefit from discussing the non-catalytic functions of PARGs in other systems. Are there any other examples for non-catalytic roles of PARGs?

23) Discussion: I think the discussion could use more detail about the work done in other systems and with PARGs. What is the direct cause of male sterility of PARP mutants? How does what is known about the defects in meiosis in PARP mutants fit into what this paper teaches us about PARG? The reduction in crossover numbers in PARP mutants can fit nicely to what is learned here about PARG.

24) Discussion line 442-449: How can PARG-1 associate with so many proteins of the synaptonemal complex (SYP-3, HTP-3, REC-8) that are positioned >50nm apart possible? Based on what is known of its size and structure can it span the distance from Axis to the middle of the central region?

25) Discussion line 451-466: The authors argue that PARG-1 doesn't act with SPO-11 since it has

much milder phenotype compared to *spo-11*. However, the same argument can be done for *HTP-3* and *MRE-11* (which are essential for DSB formation, while *RARG-1* has a much milder phenotype), yet they favor this model.

26) Discussion line 468-497: please clarify - are the changes in crossover distribution leading to both loss of crossovers on some chromosomes and to extra crossovers on others? Are all the observations regarding crossovers numbers and distribution due to the same phenomenon?

Minor comments- text

- 1) Line 70 "paternal homologs"
- 2) Line 96 "there is not gene cluster"
- 3) Line 143 "Screening"
- 4) Line 179 Figure 1C doesn't show *parg-1::GFP*
- 5) Line 367 "phospho-SUN-1 staining (Fig. S6)", phospho-SUN-1 staining is S7
- 6) Line 371 "in the *him-5*"
- 7) Line 659- the "2" in N2 is under script

Reviewer #1

The manuscript by Janisiw et al., describes the role of poly(ADP-ribose) glycohydrolase (PARG-1) in the *C. elegans* germ line. Using a combination of genetic, cell biological and biochemical approaches the authors find that PARG-1 localizes to and physically interacts with axial element and central region components of the SC, and recombination proteins, and concentrates to the short arm of the bivalent in response to crossovers. The authors provide evidence that PARG-1 functions in DSB formation, influences repair outcomes and alters the crossover landscape. Interestingly, the catalytic activity of this enzyme is not required for the reported meiotic functions, leading to a model whereby PARG-1 is functioning as a scaffold to facilitate different aspects of meiotic recombination.

In general the study is well done and will be of interest to a broad audience. However, the reader is left with a vague understanding of what PARG-1 is doing to promote these different events. This may be corrected by inclusion of additional experiments and in the writing.

1. Abstract: This is set up for elucidating the role of ADP ribosylation; however, the findings indicate that PARG-1's function in meiotic processes are independent of its catalytic activity. Please consider rewriting with less of a focus on ADP ribosylation.

Given that our study targets both the PARylation and the meiosis fields and bearing in mind that nearly nothing is known about PARylation during gametogenesis, we feel that some background should be provided, since the only known roles of PARG-1/PARG are as a PAR-degrading enzyme. We have tried our best to convey all the necessary information in the abstract without neglecting any aspects, also considering the very limited number of words available for this section.

2. In many places in the text the authors state that PARG-1 functions independently of known DSB initiation factors yet cooperates with HIM-5 (one of those initiation factors). This seems at odds to me (e.g., lines 120-121). I think all of the data are consistent with PARG-1 functioning in a parallel pathway (and perhaps not in DSB formation, see 4).

It has been recently shown that HIM-5 not only promotes DSBs but also operates in DNA repair pathway choice (Macaisne et al.; 2018). While we observe synergistic phenotypes with all the DSB initiation factor mutants (therefore indicating independent pathways), the *parg-1; him-5* double mutants instead display incomplete rescue of bivalent formation in response to exogenous DSB induction which we do not find in *parg-1; him-17*, indicating that PARG-1 does function independently of DSB initiation factors in inducing breaks yet it cooperates specifically with HIM-5 to promote repair.

Similarly, as it has been shown for *him-17*, *him-5*, *dsb-1/-2* and *xnd-1* mutants, that exogenous DSBs fully rescue loading of pro-CO factors and consequentially largely abolish formation of univalents. Together, we consider this as strong indication of defective break formation.

3. Figure 2: Please define CY, NS, CB directly on figure.

Done.

4. PARG-1's role in DSB formation: The authors provide several indirect pieces of evidence that PARG-1 plays a role in DSB formation. However, there is also evidence that it does not affect DSB formation. Although RAD-51 foci formation is delayed in the *parg-1* mutant, the numbers of RAD-51 foci are actually higher than in wild type.

We would like to emphasize here, that RAD-51 is a marker for recombination intermediates and not for DSBs directly. Of course, the former do not arise without the latter, however we believe that it is not entirely correct to infer numbers of DSBs from numbers of RAD-51 foci, especially when intermediate effects, rather than complete abrogation of DSBs, may be present.

For instance, both *him-17* hypomorphs and *him-5* mutants, although severely impaired in DSB formation, display RAD-51 foci accumulating in late pachytene, where normally no foci are observed in WT animals, indicating that reduced numbers of breaks can also influence repair dynamics.

In *parg-1* mutants, the early RAD-51 foci are severely reduced, and they peak at later stages compared to WT, suggesting that the fewer recombination intermediates formed may alter HR-repair kinetics, thereby eliciting prolonged RAD-51 loading/accumulation later on, likely caused by feedback regulation, as described in (Rosu et al.; 2013 – Stamper et al.; 2013).

What does RAD-51 look like following IR treatment in the *parg-1* mutant? Are early RAD-51 foci restored?

We have shown that consistent with delayed loading of RAD-51, abrogation of *parg-1* function also causes a delay in the early recruitment of pro-CO factors RMH-1 and MSH-5, which is restored upon irradiation (Fig.

S4). From this we can infer that also RAD-51 foci are restored at meiosis onset by exposure to IR, as otherwise the early loading of RMH-1 and MSH-5 (occurring downstream RAD-51) could not be accomplished.

Inactivation of rad-54, which traps RAD-51 on the filament, has been used as a proxy to determine total number of RAD-51-loaded breaks, examining RAD-51 in parg-1; rad-54 mutants may help clarify the role of parg-1 in DSB formation.

It has been previously proposed that in worms, DSB formation is likely to undergo a feedback regulation which ensures prolonged DSB competency when recombination is impaired (Rosu et al.; 2013 – Stamper et al.; 2013). Indeed the *rad-54* mutant has been previously employed as a tool to analyse DSB numbers by assessing RAD-51 accumulation. The fact that compromised recombination intermediate processing might also cause upregulation of break formation and consequentially increased RAD-51 foci formation might make the interpretation of the results more difficult.

We generated the *rad-54; parg-1* double mutants and analysed RAD-51 dynamics, which revealed a stark reduction in the number of foci in the *rad-54; parg-1* compared to the *rad-54* single mutants. Furthermore, we found that a large proportion of diakinesis nuclei in the double mutant showed univalents, which is never observed in the *rad-54* mutants.

However, these results still do not allow to unequivocally discriminate between the pro-DSB and the DNA repair pathway choice PARG-1-mediated activities, as while the reduction in RAD-51-labelled recombination intermediates may indeed highlight the pro-DSB function, the univalents found in the diakinesis nuclei might be a result of reduced breaks but also reflect changes in the DSB repair dynamics, therefore we preferred not to draw any strong conclusions from this data, which we have now included in Fig. S3.

Further the finding that mre-11(iow1); parg-1 mutants still have 12 univalents following IR treatment but com-1; parg-1 do not is not well explained. Interestingly, mre-11(iow1); cku-80 mutants have delayed RAD-51 foci formation while com-1; cku-80 do not, providing parallels with the parg-1 mutant. What does RAD-51 look like in the mre-11(iow1); parg-1 and com-1; parg-1 double mutants in the presence and absence of IR? What about parg-1; cku-80? At a minimum, these findings need to be discussed more to help the reader understand the complicated genetic interactions reported in the context of the published literature.

A) We assessed RAD-51 foci formation in both double mutants before and after exposure to IR, and while no differences were observed in the case of *com-1* vs *com-1; parg-1*, analysis of *parg-1; mre-11* doubles revealed the presence of several RAD-51 foci-covered nuclei before IR exposure, scattered between transition zone and mid-pachytene, which did not further increase after IR (see figure below). Importantly, this was observed only in a subset of nuclei, whereas the rest of the nuclei were not different from the *mre-11* single mutant, suggesting that the loading of RAD-51 is triggered only in some of the meiocytes. We believe that these do not represent apoptotic nuclei, as they were observed distant to the apoptotic cell death-proficient zone of the gonad (late pachytene).

We think that i) these cells may arise from pre-meiotic damage, indicating that cooperation between *parg-1* and *mre-11* could be important to preserve genome integrity during DNA replication, or that ii) PARG-1 removal from *mre-11(iow1)* could release a block to alternative, less efficient, DSB resection which could be dependent on EXO-1 (which has been shown to be active in the *mre-11; cku-80* double mutant (Yin and Smolikove, 2015)) or DNA-2. The fact that these breaks do not elicit aberrant chromatin figures in diakinesis nuclei irrespectively of exposure to IR (as *parg-1; mre-11* display mostly 12 univalents), indicates that, as observed for *rad-54; parg-1*, abrogation of PARG-1 function directs repair of recombination intermediates into a conservative mechanism which preserves chromosome morphology (e.g. sister chromatid-dependent DNA repair).

While this result is novel and intriguing, we feel that it does not add further details

regarding the roles of PARG in inducing breaks, but it rather unveils important functional and specific interactions between MRE-11-PARG-1 which would require more experiments to be clarified. Hence, we feel that this particular aspect goes far beyond the scope of our manuscript and for this reason, we decided not to include these data in the paper.

B) We built the *cku-80; parg-1* double mutant and while we did not observe variations in the early loading of RAD-51, we found a slight increase in the number of foci in late pachytene. This argues that the delayed RAD-51 loading in *parg-1* mutants is not due to illegitimate activation of NHEJ and indicates that a very minor proportion of recombination intermediates may be channelled into NHEJ repair pathway in absence of *parg-1*, which are redirected towards RAD-51-dependent HR repair once NHEJ is abolished. We have included these data in the supplementary figure S3.

5. Figure 4: Please indicate significance of the most important interactions directly on B and C.

In line with a similar request by Reviewer #3 (see below), we only left the analysis of some of the mutant backgrounds in Fig. 4 to reduce the complexity of the figure, and we moved the analysis of the controls and *parg-1* single mutants into the supplementary figure S4.

6. Figure 5: As chromosomes were not traced, please tone down the conclusion that an entire chromosome is desynapsed and therefore lacking a COSA-1 focus.

Following the reviewer's request, in the Results section (line 424) we now state: "COSA-1 foci were never associated with these unsynapsed regions" and in the Discussion (line 590) we say: "nevertheless large portions of chromatin were devoid of SYP-1/COSA-1 in many of these nuclei".

7. Figure 6: Please provide more details on how COs were examined. Were DCOs and TCOs confirmed with independent SNPs? Please provide statistical analyses of the data. I also think it is important to examine COs in the *parg-1; him-5* double mutant, the basis for uncovering the role in COs. In the materials and methods, the authors indicate that *parg-1(cd)* was generated in Hawaiian and Bristol implying that COs were analyzed, but no data on this mutant is included in Figure 6.

A set of 8 and 5 SNPs for chromosomes I and V respectively, were used to assess the recombination frequency in the indicated genetic intervals. We have now included the identification details of each of the SNPs employed in the material and methods. The DCO and TCO were consistently detected in independent biological replicates, and PCR reads were reconfirmed in each case.

We agree that it would be interesting to look at the recombination frequency in the *parg-1; him-5* double mutant, however, i) these worms are highly sterile and we would have to assess recombination in the few survivors or in the unhatched embryos, which could skew the analysis and render the result incomparable with the data already provided; ii) we have now included the analysis for the *parg-1(cd)* mutants, in which we observe no COs shift towards the middle of the chromosomes in contrast to the *parg-1* null and we did not detect a significant increase in the DCO and TCO (we only found 1 DCO). The genetic interval assessed in Ch.I is larger than in Ch. V (45cM and 35cM respectively).

We have included the statistical analysis in the supplementary Table S4, which shows significant differences between the *parg-1(gk120)* and the WT but not between the *parg-1(cd)* and the WT. This further supports that PARG-1 catalytic activity is not essential for its role in regulating meiotic recombination.

8. The authors should consider including a model, which may help them frame the discussion, and guide the reader in understanding PARG-1's role in the germ line.

We have included a new figure (Fig. 7) in which we provide a graphical rendition of PARG-1 mode of action during meiosis which blends the main results that we obtained.

Reviewer #2

I realise it is a custom to briefly summarise the main findings of the paper in a review rapport but the abstract suffices for this purpose as it is a perfect (and precise) reflection of the main findings of the study and its novelty. To me this is really an excellent paper! First of all, the technical quality of the experimental work is outstanding and state-of-the-art (an example being the generation and analysis of tagged and altered endogenous alleles), and the depth of the study is also impressive, using the full spectrum of tools available to the *C. elegans* research field. Second, I found the paper very well written, and importantly: all conclusions supported by presented data. Finally, the paper present novel biology that is relevant to the broader community: it reveals a new, physiologically very meaningful, role for PARG in meiosis, in particular, in the regulation of crossover formation, making optimal use of the fact that *C. elegans* tolerate complete lack of PARG activity. I have very little to comment or suggest to improve the quality of the paper. As a minor issue I

guess I would transfer the germline PAR staining in the *parg-1(cd)* animals from the supplement to the main figure, as I found it more convincing than (or at least importantly additive to) the Western depicted Fig 7B. We are grateful to the reviewer for all the kind comments and the deep appreciation of our work. We have moved the PAR staining in the *parg-1(cd)* mutants from the supplementary figure S7 to the main figure (Fig. 6) as requested.

I was also very much intrigued about the puzzling observation that IR could not induce the aberrant chromosome morphology typical of *com-1* and *mre-11* single mutants in the *mre-11 parg-1* double (in marked contrast to that in *com-1 parg-1* doubles), and without more experimental support or indications to what PARG is doing, I would be a bit more reserved than stating in the discussion section that this data is "indicating that PARG-1 can act as a switch in channelling DSB repair into multiple branches."

We have now provided further evidence that *parg-1* is regulating DNA repair by showing that in the *rad-54; parg-1* double mutants, a large number of univalents at diakinesis are present. This is in stark contrast to the *rad-54* single mutant, in which HR cannot progress due to blocked disengagement of RAD-51 from the chromatin, which elicits aberrant repair and the formation of ~6 misshapen DAPI bodies in the diakinesis nuclei. This was similarly observed in the *rad-54; him-5* doubles (Macaisne et al.; 2018) and reinforces our claims about PARG-1 acting in the DNA repair pathway choice.

Overall, I found the paper a great read and my compliments to the authors for such great work.

Reviewer #3

PARYlation is a protein modification used in DNA damage repair. However, little is known about it in meiosis, where DNA damage is naturally formed. Much less known about the role of PARYlation removal in meiosis. In *C. elegans*, there are two PAR glycohydrolase: PARG-1 and PARG-2. The manuscript by Janisiw et al is focused on the role PARG-1 plays in the *C. elegans* germline, as PARG-2 doesn't seem to play a role in the germline under normal conditions. PARG-1 localizes, and physically interacts, with the synaptonemal complex. *parg-1* mutants show a delay and a small accumulation of RAD-51 foci. Although crossover numbers are not affected in the *parg-1* mutant, the *parg-1* mutant act synergistically with mutants with partial defects in crossover formation and suppress defects in recombination of mutants that target repair to NHEJ. Most of the defects observed in the double mutants can be suppressed by introducing DSBs using irradiation. These findings are interpreted as a role for PARG-1 in DSB formation. The contribution of PARG-1 (by its own) is small - embryonic viability is not much effected and the obligatory crossovers still form. All of these phenotypes are independent of the catalytic activity of PARG-1. The *parg-1* mutant also shows altered distribution of crossovers and decrease in interference, indicating additional function for PARG-1 in crossover regulation. Overall, the discovery that PARG-1 plays a role in meiotic recombination and that this function is independent of its catalytic domain is novel and exciting. The analysis of the *parg-1* mutant phenotypes is very nicely and thoroughly done and the quality of the imaging performed is excellent. If the conclusion stated in this paper will be supported by addressing the concerns below, it will likely provide findings that will be of interest to the community.

1) In *parg-1* mutants RAD-51 foci delay in their appearance and peak later than in wild type. *parg-1* mutants can enhance the phenotype of mutants that affect the formation of DSBs, and this effect can be partially suppressed by irradiation. These observations can be interpreted in two ways: 1) PARG-1 has an effect on the timing of DSB formation: in its absence DSB formation is delayed, 2) PARG-1 is promoting both DSB formation and DSB repair: in its absence less DSBs are formed but the DSBs formed take longer to repair. The authors favor option #2. However, option #1 is just as likely and is more parsimonious. DSB formed later (compared to wild type) may be more sensitive to perturbation of the DSB formation machinery explaining their synergistic effect with mutants such as *him-17* or *dsb-2*. DSB formed later may be formed in a germline region that is less permissive to NHEJ, explaining why *parg-1* can suppress chromosomal fusions in *com-1* or *mre-11* mutants. The partial suppression by IR can be explained by the timing of IR breaks: IR breaks are induced throughout the germline and likely the diakinesis oocytes scored originated from brakes in the distal germline. Therefore, by exposing to IR breaks are added before where they are made in *parg-1* mutants, suppressing the DSB timing defects. Unless there is a very good reason to disregard model #1 the papers should be written in a way that accepts both possibilities equally, including the discussion, title and abstract. We agree with the Reviewer in that both models may be right and without a direct quantification of DSBs, which is at the moment technically not possible in *C. elegans*, neither of them can be entirely ruled out. However, our genetic analysis favours the second option since we think that delayed DSB formation only, coupled with a possible perturbation in the repair pathways (option #1), would most likely not prevent

chiasmata formation. It has been extensively shown (and we observe the same) that irradiation of all pro-DSB factor mutants elicits full rescue of bivalent formation in the diakinesis nuclei observed 20h-27h post IR. These diakinesis nuclei were at the mid pachytene stage at the moment of irradiation, indicating that the DSBs formed in later stages during meiotic progression are nonetheless fully proficient in forming chiasmata, and therefore the possibility that only a delayed formation of DSBs may be at the base of the phenotypes observed upon *parg-1* removal is less likely.

Our analysis in the diakinesis nuclei of irradiated animals was conducted 27h post-IR as in most of the previous studies, meaning that these cells were at the mid-pachytene stage at the moment of irradiation rather than at the distal tip, as it has been shown that nuclei residing in the distal portion of the gonad take 54-60 hours to reach the diakinesis stage (Jaramillo-Lambert et al.; 2007).

The fact that these breaks, however, might somehow alter repair dynamics is entirely possible and we actually believe that this may be the case, since we already claim that PARG-1 promotes both formation and repair of meiotic DSBs. We have nonetheless tried to better convey the possibility that also perturbed kinetics in the induction/repair of DSBs may be a possible cause of the phenotypes that we observe, by changing the title and modifying the abstract and the discussion as the Reviewer suggested.

2) I find the fact that the catalytically dead mutants share no phenotypes with the null mutant interesting but also concerning. A major takeaway message of this paper is that PARGs can play a non-catalytic role in DSB formation and CO regulation. Since this is such an unexpected result, it needs to be reinforced. Both null and CD mutations increase PAR, therefore they are targeting PARG-1, but it is possible that an additional mutation in the null background is responsible for the null phenotypes (even if this allele was outcrossed X6, linked mutations may still be present). It will be better to confirm some basic phenotypes of the null with more than one allele. CRISPR/Cas9 is widely used in *C. elegans* and in the hands of the authors and deletion alleles can be generated in a span of few weeks. It will be advisable to generate a full null allele of *parg-1*. There is no need to examine all phenotypes with this allele, but to examine the phenotypes that are different between the CD and the null alleles (EMB, anti-RAD-51, and if possible, the interaction with *him-5*).

We have generated a strain carrying the same identical deletion present in VC130 (*parg-1(gk120)*) by CRISPR (*parg-1(DDR51)*) that fully recapitulates the phenotypes of the original *parg-1(gk120)* mutant allele (EMB, RAD-51 accumulation, and interaction with *him-5* as suggested by the Reviewer), demonstrating that the phenotypes observed, result from impaired *parg-1* function and are not due to secondary mutations. We have included the embryonic lethality data in Fig. S3G, the RAD-51 analysis in Fig. S3C and the diakinesis phenotype in Fig. 4D, together with the quantifications done for the *parg-1(gk120)* alone and in combination with the *him-5* mutant.

3) Statistical analysis is missing for much of the data. In some cases, there are error bars but not statistics. It is essential to perform this analysis (Fig 1B, 4B, 3A, 4B, 4C, 4D, 5A, 5B, 6A, 6B, 7C, 7E, 2SB, S3B, S3C, S5A, and S7A).

We have included the statistical analysis as requested.

4) Others (Bae et al 2020 FASEB letters) have shown a role for PARG-2 in the germline in response to IR. Here the paper shows that PARG-2 plays no role in the germline, but this is done in the absence of IR. One way to reconcile both observations is for *parg-2* gene/PARG activity to be induced by IR. If so, PARG-2 expression could compensate for the lack of *parg-1* activity in IR experiments, suppressing the defects observed in the absence of IR. In this case, the suppression of the phenotype of double mutant (with *parg-1*) by IR can be attributed to PARG-2 activation. This should be tested (for example, by staining for PAR in germline of *parg-1* mutants a few hours after IR - is PAR staining weaker compared with no IR?).

PAR staining is extremely bright also under physiological conditions of growth in the *parg-1* mutants and therefore exposing the worms to IR and perform PAR staining would not significantly tilt the balance.

Bae et al., 2020 show that there are no additive phenotypes in the *parg-1^{RNAi}parg-2(ok980)* worms (or *parg-1(gk120)parg-2^{RNAi}* animals), suggesting a common pathway for repair of exogenous DSBs, however the fact that i) absence of *parg-1* accumulates PAR while abrogation of PARG-2 function does not, ii) the *parg-1; him-5* double mutants show a synergistic phenotype whereas the *parg-2; him-5* mutants do not, indicates that these two paralogs exert separable functions.

To address a possible role exerted by PARG-2 in response to IR, we generated *parg-1 parg-2; him-5* triple mutants and analysed diakinesis nuclei before and after IR.

The analysis in the non-irradiated animals shows that diakinesis nuclei in the *parg-1 parg-2; him-5* triple mutants have the same level of achiasmatic chromosomes as the *parg-1; him-5* double mutant in stark contrast to the *parg-2; him-5* mutant, in which only the chromosome X is achiasmatic (recapitulating the same phenotype of *him-5* single mutants as already shown in Fig. S6). Exposure to IR restored bivalent formation in the *parg-1 parg-2; him-5* as similarly observed for *parg-1; him-5*, further confirming that *parg-1*, but not *parg-*

2, plays essential roles in regulating HR-mediated repair in absence of HIM-5. We included these data in Fig. S6.

The discrepancies between our data and those by the Koo's lab could be ascribable to i) different alleles, as our *parg-2* allele is a full knock-out whereas the *parg-2(ok980)* employed in the manuscript by Bae et al.; is a partial deletion and this could give rise to gain of function phenotypes; ii) the IR dose that we used (10 Gy) is much lower than theirs (45-70-80 Gy).

It is also worth mentioning that the original study where *C. elegans* PARG-1 and PARG-2 (then called PME-3 and PME-4) were identified and characterised (St-Laurent et al.; 2007) showed that PARG-1 is predominantly expressed whereas PARG-2 expression levels are barely detectable; further, the authors showed that while PARG-1 was mostly nuclear, PARG-2 accumulated predominantly in the cytosol. Therefore, while belonging to the same class of enzymes, the two proteins are certainly differently regulated both in terms of expression levels and tissue specificity.

5) Most of the figures are written in a small font that makes it hard to examine them. The font size varies between panels. Using a larger and standard font size can improve the figures.

We have corrected font variations and increased the font size where possible.

6) The western blot showing that there is no PARG-1 protein in the mutant is shown just for 100kDa. However, if residual protein was made it should have been smaller (since the mutant contains large deletion), so the WB that includes the expected size following this deletion should also be shown. The fact that *gk120* is not a complete deletion opens the door for gain of function/hypomorphic phenotypes, so it's important to completely rule this out.

We have now provided the picture of a full membrane with WT, *parg-1(gk120)* and *parg-1(DDR51)* total extracts probed with anti PARG-1 antibody, which confirms that these are null mutants (Fig. S1B). We detect some identical bands below 50 kDa in all three extracts, however there are no predicted isoforms corresponding to this size, suggesting cross-reaction products of the antibody.

7) What is the localization pattern of PARG-1 in diakinesis? The data we see is from late pachytene, but a clearer image should be observed in diakinesis.

We have included a picture of a diakinesis nucleus from the *parg-1::GFP* strain (Fig. S1C). At this stage, the GFP signal is very bright throughout the nucleus, although some signal together with the chromosomes is also visible. However, given the high intensity of the protein in the nucleoplasm (even throughout the z stacks) it is hard to say whether this signal is truly bound to chromosomes and therefore we did not emphasize it.

8) It is a little bit perplexing that PARG-1 co-IPs with every single protein tested. All these IPs use a GFP tagged protein. Could that create a problem?

We have conducted co-immunoprecipitation experiments by employing a strain carrying an integrated single GFP copy on chromosome II with ubiquitous expression, which we previously employed to perform a similar control (Janisiw et al.; 2018, Plos Genetics). We immunoprecipitated the GFP and performed western blot to reveal PARG-1. As shown in the figure on the right, in this case PARG-1 was not pulled-down with the GFP, indicating that the interactions observed in our IPs are indeed specific.

Moreover, we would like to mention that we have previously performed Mass Spectrometry analysis on GFP pull-downs from PARG-1::GFP and untagged WT backgrounds, in which we found many axial components (i.e. HTP-1/-2/-3, HIM-3, COH-3/COH-4, REC-8 and SMC-3/-4/-5), cohesin subunits SCC-1 and SCC-3, synaptonemal complex component SYP-1, and several recombination factors. Although all of these interactors were specifically enriched in the pull-downs of the PARG-1::GFP but not of the WT, we did not have the possibility of repeating the experiment in order to obtain a biological replicate, therefore we felt that only one trial was not sufficient and decided not to include these data in the manuscript. However, we thought this could be informative in the light of the Reviewer's concern, as the co-immunoprecipitations shown in Fig. 4-5 recapitulate some of the same interactions observed by MS.

9) I find it confusing that the same type of data is presented in different ways. For example, could the data pending DAPI body number (3C, 4D, 7F), embryonic lethality (1B, 2C) presented in the same way?

Following the Reviewer's request and also in line with *Nature Communication's* formatting requirements, we have modified most of the charts displaying distributions with scatter plots instead of bars, except for the RAD-51 data.

10) *How many repeats/replicas were there for the western blot/IP data?*

All the biochemistry experiments were performed in biological duplicates.

11) *PARG-1 in 2C runs at 100KDa, while in other westerns it runs higher- is this a typo?*

Since there is no Western blot in 2C, we suppose that the Reviewer meant 2B. In 2B-left, untagged endogenous PARG-1 is detected in the wildtype strain whereas in 2B-right, the PARG-1::GFP fusion protein is detected with both anti-PARG-1 or anti-GFP antibodies, revealing a higher mW as expected. In the co-IPs in 2I-2J, the detected endogenous PARG-1 runs indeed similarly as in 2B (slightly above the 100 kDa band of the ladder).

12) *Line 173 states that PARG-1 is expressed in all nuclear compartments, but the figure legend says it is enriched in the nucleus, which fits the figure better. Which one is right and how reproducible?*

In the previous version of the manuscript, line 173 stated "We find expression of PARG-1 in all subcellular compartments in wild-type animals (Fig. 2A)" and not "nuclear compartments", as the protein is in fact detected in both nuclear and cytosolic (subcellular compartments) extracts. In the figure legend we said "Western blot analysis of fractionated extracts detects PARG-1 in all subcellular compartments with enrichment in the chromatin-bound fraction". Therefore, we do not find inconsistencies between these statements and what is shown in the figure and the results have been reproducibly obtained in both of the performed replicates. However, in the revised version of the paper we now state in line 173 "We found expression of PARG-1 in both the cytosol and the nucleus in wild-type animals" in order to increase clarity.

13) *Based on 2G, It's hard to see that the localization is in the nucleoplasm. It looks like it's still associated with chromosomes, but since the chromosomes are not synapsed it doesn't look like wild type localization. A zoom-in with an arrow showing GFP where there is no DAPI will be helpful.*

Chromosomes are not synapsed in the *syp-2* mutant either (Fig. 2H) however PARG-1::GFP localization looks clearly different (associated with the axes) compared to *htp-3* mutants.

We have included the overlay with the DAPI channel which shows no GFP-tracks clearly associated with the DNA (indicated by the arrowheads) but rather a punctate staining scattered across the chromatin.

14) *2I- why 2 bands with HTP-3?*

Both bands are specific, as no signal is detected in the untagged WT worms. They could originate from either an alternative HTP-3 isoform, post-translational modifications or possibly a degradation product, since large GFP-tagged proteins (as HTP-3) are known to often undergo this phenomenon.

15) *4B and 4C are very hard to read. I suggest focusing on one or two zones and showing the rest in sup figure.*

See response to point 5 of Reviewer #1.

16) *5A and B- why are there less COSA-1 foci and more desynapsis in *parg-1* single mutant following IR? Statistics will help here...*

Statistical analysis revealed non-significant differences between the control and the *parg-1* mutants after IR with respect to COSA-1 foci, however the extent of de-synapsis is indeed statistically significant, and we have indeed reproducibly observed nuclei with partial synapsis in the *parg-1* mutants especially after IR. Previous work in worms (Couteau and Zetka, *Developmental Cell*, 2011) has shown that chromosome axes in late pachytene nuclei can undergo a certain degree of separation upon exogenous damage, coupled with a reversible loss of H2AK5Ac and local desynapsis. This behaviour, while quite rare in the WT, seems to be exacerbated in the presence of mutated HTP-3 (which still localizes to axes) or upon abrogation of ATM-1 and MRE-11 function, suggesting that while not required to elicit axes separation, proper DNA repair is important to limit the extent of separation. Given its interaction with HTP-3-MRE-11 and localization to the chromosome axes, it could be that loss of PARG-1 somehow mimics this destabilization of the SC resulting in occasional desynapsis. However, this is purely speculative, as further analysis would need to be done to address the roles of PARG-1 along the SC.

17) *Figure 5 and 6 can be fused to one figure.*

Done.

18) If a point wants to be made about increased localization in CD mutants, 7A should be quantified (like in 2D).

We have now included insets of non-deconvolved nuclei from late pachytene stained with GFP and SYP-1 in Fig. 6C (as in 2D), which show that while in the controls PARG-1::GFP starts already to retract toward the short arm of the bivalent (as also indicated by the bright GFP agglomerates), the PARG-1(CD)::GFP is still localizing in longer tracks and the signal is brighter, suggesting increased protein levels which were indeed confirmed by the western blot analysis (6C).

19) Line 500-503 "parg-1(cd) mutants...displayed delayed redistribution along the chromosomes in late pachytene" I cannot see that.

See above.

20) Line 703-707: indicate which data was collected from 2 vs. 3 germlines. 2 germlines may be too little for some readers.

Two germlines were scored only for the WT (which do not have any PAR signal) and the *parp-1; parg-1*, which consistently give a reduced staining intensity. We quantified ~100 nuclei (and more) in the mutant backgrounds analysed (see Methods), which is a commonly used number in the field.

21) Image 1A- please include scale bar in bp. It will also be useful to have a cartoon of the proteins with domain structure, what the deletions remove and where is the CD mutations. This is important since the catalytic domain is examined. It will also be informative to discuss the structure of PARG-1 and 2 and their similarities to PARGs found in other organisms, which will be important for readers outside the *C. elegans* field.

We have replaced the cartoon in Fig. 1A with a new one, depicting all the required details about the mutant alleles and the protein domains. We have also included more details in the discussion as the Reviewer suggested.

22) Discussion: the discussion could benefit from discussing the non-catalytic functions of PARGs in other systems. Are there any other examples for non-catalytic roles of PARGs?

At best of our knowledge, the roles of PARG during meiosis (catalytic or non-catalytic) have never been investigated so far and as such, we cannot draw any parallels with other species. However, we cited the study of Kaufmann et al., in which the authors show that PARG interacts with PCNA through a non-canonical PIP-box and that a mutation of a residue (K409) within the PIP-box abrogates its interaction with PCNA, impairs PARG localization within replication foci and PARG recruitment to DNA damage sites, but does not affect PARG catalytic activity. Mutation of the PARG catalytic site also has no effect on PARG-PCNA interaction. These data are not directly comparable with our findings, since the work was done in mammalian mitotic cells, however they do emphasize that PARG holds other functions beyond its enzymatic activity in PAR degradation, which have yet to be fully explored.

23) Discussion: I think the discussion could use more detail about the work done in other systems and with PARGs. What is the direct cause of male sterility of PARP mutants? How does what is known about the defects in meiosis in PARP mutants fit into what this paper teaches us about PARG? The reduction in crossover numbers in PARP mutants can fit nicely to what is learned here about PARG.

Two previous studies analysing *Parp1* or *Parp2* deficient mice, which we have already cited, revealed defects in DNA repair and CO formation, as well as extensive cell death by apoptosis during spermatogenesis (Yang et al.; 2009; Dantzer et al.; 2006). However, since these two genes display a partial redundancy in vertebrates, a direct analysis in a background lacking PARPs altogether has not been done, possibly indicating that more severe phenotypes may exist in their absence during germ cells development. We have collected some data on the *C. elegans parg-1; parp-2* double mutants, which we generated in the lab. They do not reveal a direct role in CO formation or in physiological DSB repair, suggesting that PARylation is largely dispensable for proper completion of meiotic Prophase I in nematodes. We cannot rule out that other PAR polymerases may be active in worms and somehow perform additional functions in absence of PARP-1 or PARP-2, such as Tankyrases (worms have one ortholog), although it has been shown in mammals that they cannot compensate for lack of *Parp1* or *Parp2*.

24) Discussion line 442-449: How can PARG-1 associate with so many proteins of the synaptonemal complex (SYP-3, HTP-3, REC-8) that are positioned >50nm apart possible? Based on what is known of its size and structure can it span the distance from Axis to the middle of the central region?

One possibility could be that there are different coexisting pools of PARG-1 (i.e. along the axis and on the central elements of the SC), which may explain interaction with these different factors. This is supported by

the finding that in the *syp-2* mutant (Fig. 2G), PARG-1 must be loaded along chromosome axes (as there is no synapsis). Moreover, PARG retraction towards the short arm of the bivalent suggests that it can also associate with central elements of the SC.

Another possibility could be what the Reviewer is suggesting and PARG-1 might form a bridge between lateral and central elements. It is difficult to infer from its physical properties, as its localization during mammalian meiosis is not known and there is no information on the structure of *C. elegans* PARG-1. We have previously tried to perform super-resolution analysis via SIM microscopy to gain more information on its localization, but unfortunately PARG-1::GFP does not withstand the fixation protocols required for nuclear spread preparation and becomes no longer detectable (with or without employing anti-GFP antibodies).

25) Discussion line 451-466: The authors argue that PARG-1 doesn't act with SPO-11 since it has much milder phenotype compared to *spo-11*. However, the same argument can be done for HTP-3 and MRE-11 (which are essential for DSB formation, while PARG-1 has a much milder phenotype), yet they favor this model.

We have removed this sentence from the discussion.

26) Discussion line 468-497: please clarify - are the changes in crossover distribution leading to both loss of crossovers on some chromosomes and to extra crossovers on others? Are all the observations regarding crossovers numbers and distribution due to the same phenomenon?

With the available data, we cannot distinguish between these two possibilities. The aberrant recombination landscape and the reduced COs observed along Ch. I and Ch. V does not allow to deduce whether or not these extra COs along these two chromosomes are coupled with lack of them on others. In the *parg-1* single mutants this is not the case, as we don't find achiasmatic chromosomes in diakinesis, indicating that the supernumerary COs do not prevent formation of the obligate CO elsewhere in the genome and without a marker for Class II COs we cannot monitor these events cytologically.

It is worth mentioning though, that many of the mutants with reduced DSBs or reduced CO formation, such as *him-5*, *rec-1* and *rmh-1*, display an increased recombination in the centre of the chromosomes. Also *rtel-1* and *dpy-28* mutants, known to have a several folds increase in the "class II" COs, are fully competent in forming the six obligate, COSA-1/MSH-5/ZHP-3-labelled COs, suggesting that the two phenomena are indeed not incompatible.

Minor comments- text

1) Line 70 "paternal homologs"

Line 70 states "Connected parental homologous chromosomes" not "paternal", therefore we left it as is.

2) Line 96 "there is not gene cluster"

We have removed this sentence.

3) Line 143 "Screening"

We have rephrased by saying "assessment".

4) Line 179 Figure 1C doesn't show *parg-1::GFP*

We have corrected this.

5) Line 367 "phospho-SUN-1 staining (Fig. S6)", phospho-SUN-1 staining is S7

We have corrected this.

6) Line 371 "in the *him-5*"

We have corrected this with "*him-5* mutants".

7) Line 659- the "2" in N2 is under script

We have corrected this.

Reviewers' Comments:

Reviewer #1:

Remarks to the Author:

The revised manuscript by Janisiw et al., describes the role of poly(ADP-ribose) glycohydrolase (PARG-1) in the *C. elegans* germ line. Using a combination of genetic, cell biological and biochemical approaches the authors find that PARG-1 localizes to and physically interacts with axial element and central region components of the SC, and recombination proteins, and concentrates to the short arm of the bivalent in response to crossovers. The authors provide evidence that PARG-1 functions in DSB formation, influences repair outcomes and alters the crossover landscape. Interestingly, the catalytic activity of this enzyme is not required for the reported meiotic functions, leading to a model whereby PARG-1 is functioning as a scaffold to facilitate different aspects of meiotic recombination. The authors have done an excellent job addressing the previous reviews. I just found a couple of minor things:

1. On line 259, I recommend removing "minor"
2. On line 321, "take" should be "took"

Reviewer #2:

Remarks to the Author:

I have no further comments, and support publication of this work in Nature communications.

Reviewer #3:

Remarks to the Author:

The authors provided a comprehensive and logical rebuttal and addressed all my concerns both experimentally and by modifying the text. I see no further issues with this manuscript.

Reviewer #1

The revised manuscript by Janisiw et al., describes the role of poly(ADP-ribose) glycohydrolase (PARG-1) in the C. elegans germ line. Using a combination of genetic, cell biological and biochemical approaches the authors find that PARG-1 localizes to and physically interacts with axial element and central region components of the SC, and recombination proteins, and concentrates to the short arm of the bivalent in response to crossovers. The authors provide evidence that PARG-1 functions in DSB formation, influences repair outcomes and alters the crossover landscape. Interestingly, the catalytic activity of this enzyme is not required for the reported meiotic functions, leading to a model whereby PARG-1 is functioning as a scaffold to facilitate different aspects of meiotic recombination. The authors have done an excellent job addressing the previous reviews. I just found a couple of minor things:

- 1. On line 259, I recommend removing “minor”*
- 2. On line 321, “take” should be “took”*

Done**Reviewer #2**

I have no further comments, and support publication of this work in Nature communications.

Reviewer #3

The authors provided a comprehensive and logical rebuttal and addressed all my concerns both experimentally and by modifying the text. I see no further issues with this manuscript.